# Light fermions in color: why the quark mass is not the Planck mass

**Gustavo P. de Brito** [iD]$^a$ **Astrid Eichhorn** [iD],$^a$ **Shouryya Ray** [iD]$^a$

$^a$*CP3-Origins, University of Southern Denmark, Campusvej 55, DK-5230 Odense M, Denmark*

*E-mail:* gustavo@cp3.sdu.dk, eichhorn@cp3.sdu.dk, sray@cp3.sdu.dk

ABSTRACT: We investigate whether quantum gravity fluctuations can break chiral symmetry for fermions that are charged under a $U(1)$ and an $SU(N_c)$ gauge symmetry and thus closely resemble Standard-Model fermions. Unbroken chiral symmetry in the quantum-gravity regime is a necessary prerequisite to recover the Standard Model from a joint gravity-matter theory; if chiral symmetry is broken by quantum gravity, fermions cannot generically be much lighter than the Planck mass and the theory is ruled out. To answer this, we work in a Fierz-complete basis of four-fermion interactions and explore whether they are driven to criticality.

We discover that the interplay of quantum gravity with the non-Abelian gauge theory results in chiral symmetry breaking, because gravitational and gauge field fluctuations act together to produce bound states.

Chiral symmetry breaking is triggered by four-fermion channels that first appear when non-Abelian charges are introduced and that become critical if the non-Abelian symmetry is gauged. Extrapolating our result to the Standard Model fermions, we conjecture that the non-Abelian gauge coupling, Abelian gauge coupling and Newton coupling are all bounded from above, if Standard Model fermions are to remain much lighter than the Planck mass.

In contrast, fermions that are charged under a global non-Abelian group can remain light for arbitrarily large values of the Newton coupling. We find that different chiral symmetries are emergent at low energies, depending on the strength of the gravitational coupling. This is an example of fixed points with different degrees of enhanced global symmetry trading stability in fixed-point collisions driven by gravitational fluctuations.

## 1   Introduction

One of the most profound phenomena in physics is arguably a phase transition, where a system changes some or all of its key characteristics, typically while breaking a symmetry. In high-energy physics, this is particularly important, because symmetries protect certain properties of fields and their interactions. As a fundamentally important example, chiral symmetries, under which the left- and right-handed components of Dirac fermions transform separately, protect fermions from acquiring masses. Chiral symmetry breaking is therefore tied to fermion mass generation. In the Standard Model (SM), the most prominent example is chiral symmetry breaking in QCD, which is responsible for the bulk of

the hadron masses.[1] Chiral symmetry breaking can be monitored in four-fermion interactions. If these become critical, i.e., of diverging interaction strength, fermion bilinears form condensates and associated bound states. In turn, the interaction between fermions and the condensates renders fermions massive, because chiral symmetry is no longer intact to protect the lightness of fermions.

The study of four-fermion interactions therefore has a long tradition in condensed-matter systems [1, 2]. Four-fermion interactions have also been in focus in QCD and beyond, see [3] for a review. In QCD, four-fermion interactions can be used to detect the onset of chiral symmetry breaking, see, e.g., [4, 5] which is responsible for the bulk of the baryon mass. Beyond QCD, four-fermion interactions play a key role, when the Higgs field is substituted by an emergent degree of freedom, e.g., in technicolor models [6] and similar settings, see, e.g., [7].

Finally, four-fermion interactions have come into focus in (asymptotically safe) quantum gravity, because the attractive nature of gravity was expected to favor the formation of bound states and the associated chiral symmetry breaking. In [8], see also [9–11], it was found that this expectation is not realized: while gravity as a classical force is attractive, quantum fluctuations of gravity do not favor chiral symmetry breaking. As a consequence, fermion mass generation at the Planck scale is avoided.[2] In turn, these results pave the way for an asymptotically safe SM with gravity, which is only viable if fermions remain light all the way down to the electroweak scale. Recently, it was found that a similar result holds in quadratic gravity [13]. Whether or not unbroken chiral symmetry for fermions may even be a universal property of all (metric) QFTs of gravity is an intriguing open question. Further, four-fermion interactions also provide information on the proton life time, an observable that can (nearly) constrain Planck-scale physics, see [14].

The situation becomes more interesting if fermions couple not only to gravity (as they unavoidably do), but also to a gauge field. Quantum fluctuations of gauge fields, unlike those of gravity, trigger chiral symmetry breaking if they are sufficiently strong [4, 15].

Taken together, gravity and Abelian gauge interactions compete; and, at an asymptotically safe fixed point, this competition results in a lower bound on the number of fermions [11].[3]

A key open question is, therefore, whether fermions that are charged under a non-Abelian gauge symmetry remain light in quantum gravity. We thus study the effect of quantum-gravity fluctuations on four-fermion couplings in a setting in which the fermions are also charged under an Abelian and a non-Abelian gauge group. This study has several motivations:

---

[1]Chiral symmetry breaking through the Higgs mechanism is negligible for the masses of hadrons, but matters for the mass difference between proton and neutron, resulting in the instability of free neutrons and stability of free protons.

[2]An explicit Dirac fermion mass term is a relevant perturbation of an asymptotically safe fixed point [12], but also an explicit source of chiral symmetry breaking.

[3]An upper bound on the number of fermions arises in order to avoid gravitational catalysis [16, 17], i.e., the breaking of chiral symmetry through a negatively curved background spacetime. Gravitational catalysis may operate at small scales even in a universe that is positively curved at large scales and thus these bounds may be relevant in our universe.

a) Most importantly, our study addresses the open question whether quarks, which carry both color and Abelian gauge charge, can remain light in quantum gravity.

b) Furthermore, it paves the way for future work on (extended) technicolor or more general composite Higgs models and their embedding into quantum gravity. In (extended) technicolor models, four-fermion interactions must become critical at a sufficiently high scale to produce a phenomenology that resembles the SM Higgs sector sufficiently to be compatible with present data. It is thus an intriguing open question whether the associated symmetry-breaking patterns can be accommodated, given the constraints on four-fermion couplings in quantum gravity.

c) Additionally, our study is motivated by models of dark matter. Because models of scalar dark matter are highly constrained in asymptotically safe quantum gravity [18–22], with the arguably simplest model already ruled out [23], it is of interest to explore whether fermionic dark matter is a viable candidate. Dark sectors with strong dynamics [24–27] have not been studied in asymptotically safe gravity so far, and our study paves the way for future work in this direction.

d) At a more formal level, we investigate a system which exhibits two different types of fixed-point collision as a function of gravitational coupling strength: For a global non-Abelian symmetry, fixed points collide and remain real, realizing an example of stability-trading. As a consequence, different chiral symmetries are emergent for different values of the gravitational coupling strength. For a local non-Abelian symmetry, fixed points collide and subsequently become complex, realizing what has been referred to as the weak-gravity bound in the literature [8, 10, 28–33]. As a consequence, the gravitational coupling strength is bounded from above.

The remainder of this paper is organized as follows: In Sec. 2, we review the treatment of gravity as a quantum field theory in general, and its completion using quantum scale symmetry within asymptotic safety in particular. We then review, in Sec. 3, the technical framework of the functional renormalization group (FRG) we use to derive the RG flow of four-fermion couplings. We record these in Sec. 4. Finally, we review the different known scenarios for fixed-point collisions and their physical consequences in Sec. 5. In Sec. 6, we place what is known about chiral symmetry breaking in fermion systems coupled to gravity within this classification scheme for fixed-point collision scenarios. With this background material at hand, we turn our attention to the new fixed-point structure of colored and charged fermions. In Sec. 7, we proceed to discuss chiral symmetry of fermions with global color charge (i.e., when all gauge couplings vanish) coupled to gravity. It turns out this setting already accommodates enough structure to feature non-trivial fixed-point collision scenarios, viz. in terms of the symmetry realized at the chirally symmetric fixed point. We then turn our attention to the full interacting system, where both non-Abelian and Abelian gauge fluctuations appear, in Sec. 8. We elucidate the different chiral symmetry breaking mechanisms at work (Sec. 8.1) and discuss resulting bounds on the non-Abelian gauge coupling, as well as potential restrictions on the number of gluons and fermion flavors

(Secs. 8.2–8.4). We also discuss, for the first time known to our knowledge, a gravity-induced asymptotically safe UV completion for the non-Abelian gauge sector, if the number of fermion flavors dominates the number of gluons, in Sec. 8.5. We close by summarizing our findings and commenting briefly on directions for future research in Sec. 9.

## 2 Background: Gravity as a quantum field theory

Einstein gravity is perturbatively nonrenormalizable in four spacetime dimensions. This technical result has important physical consequences, in that it signals that predictivity breaks down. This breakdown is due to a proliferation of counterterms at increasing orders in the loop expansion. Each counterterm comes with a coupling that is a free parameter of the theory.

Predictivity may be achieved if a symmetry is imposed on the theory. Traditionally, symmetries are imposed at the classical level and one then hopes that quantization can proceed in an anomaly-free way. Supergravity is an example, in which the supersymmetry indeed reduces the number of counterterms dramatically. Yet in four dimensions it is not expected to suffice to restore predictivity (cf. [34] and references therein).

Asymptotic safety, or quantum scale symmetry, is a distinct form of symmetry. It is not imposed at the classical level, instead, it only emerges at the quantum level. It is realized, if the dimensionless counterparts of the scale-dependent couplings are constant at high energies. This can only be achieved, if quantum fluctuations either balance each other (e.g., screening and antiscreening contributions of different fields to a given coupling) or balance the canonical scaling of couplings. Thus, asymptotic safety is a form of scale symmetry that emerges at the quantum level only. Technically, it is realized at an interacting ultraviolet (UV) fixed point of the Renormalization Group (RG).

Asymptotic safety can restore predictivity to the theory, if it comes with a finite number of relevant directions, i.e., couplings along which the RG flow can depart from the quantum scale symmetric regime. Each relevant direction gives rise to a free parameter of the theory. All other couplings – also associated to free parameters in the perturbative quantization – can be calculated as functions of these free parameters.

There is by now compelling theoretical evidence for asymptotic safety in gravity, see [35–39] for recent reviews, [40, 41] for textbooks, [42, 43] for lecture notes, [44] for a critical discussion of the status and, e.g., [45–83] for key papers.

Additional support also comes from lattice simulations which find a second-order phase transition in the quantum gravitational phase diagram [84]. A higher-order transition is a necessary prerequisite for asymptotic safety. There is also good evidence that asymptotically safe gravity has three relevant directions, see, e.g., [53, 62, 70, 85].

Going beyond the pure-gravity theory, there is also promising evidence that the Standard Model of particle physics, ameliorated by gravity, becomes asymptotically safe, see [86, 87] for recent reviews of the field and [88] for an introductory perspective. Under the impact of quantum fluctuations of Standard Model matter, gravity remains asymptotically safe [89–94]. The interplay of asymptotic safety with scalars [31, 32, 79, 89–91, 94–103], fermions [9, 12, 79, 89, 90, 93, 94, 104–109] and gauge fields [79, 89, 91–94] has also been

investigated separately. There is evidence that some of the couplings of the Standard Model can either be calculated from first principles or bounded from above due to the predictive power of quantum scale symmetry [10, 29, 110–115], see also [116, 117]. Similar predictive power is present beyond the Standard Model, see [18–23, 118–128] for examples.

In addition, the higher-order Wilson coefficients of the Standard Model effective field theory [129] are also constrained by asymptotic safety. Typically, these higher-order couplings are suppressed by a power of the Planck scale, due to their irrelevant character at the asymptotically safe fixed point. Therefore, they are not likely to be relevant for particle-physics phenomenology at low energy scales. Nevertheless, they can play an important role for the physics beyond the Planck scale, because strong enough gravitational fluctuations may either prevent a fixed point in these couplings or render them relevant, introducing new free parameters in the Standard Model [10, 28–31, 33, 103]. One particularly important set of higher-order couplings are the four-fermion couplings. These contain information on the formation of bound states and the associated breaking of chiral symmetry. In other words, four-fermion couplings are decisive for the mass of fermions and therefore their study indicates whether or not quantum gravity is compatible with fermions being much lighter than the Planck mass.[4] Starting from [8], see also [9], where no indication for chiral symmetry breaking was found in a gravity-matter system, [10] investigated the same system in the presence of a Yukawa coupling and [11] in the presence of an Abelian gauge coupling. Here, we add a non-Abelian gauge coupling for the first time and discover that it makes an important difference in the dynamics of the system.

## 3 Technical setup

In this section, we provide a short review of the functional renormalization group (FRG), which we use to compute the running of four-fermion couplings. For more details on the FRG see, e.g., [132] for a general review and [36, 38, 39, 43] for introductions and reviews in the quantum gravitational context. The FRG is based on the flowing action $\Gamma_k$, a functional that describes the effective dynamics of (Euclidean) systems, which one obtains after integrating out modes characterized by a momentum-scale $p \equiv (p \cdot p)^{1/2}$ larger than $k$. One uses $k$ to denote the renormalization group scale that separates IR from UV modes. In the FRG, one introduces $k$ as part of a IR-regulator function $\mathbf{R}_k(p^2)$ added to the (Euclidean) generating function as a type of momentum-dependent mass term.

The flowing action $\Gamma_k$ satisfies a formally exact flow equation

$$k\partial_k \Gamma_k = \frac{1}{2}\mathrm{STr}\left[\left(\Gamma_k^{(2)} + \mathbf{R}_k\right)^{-1} k\partial_k \mathbf{R}_k\right]. \tag{3.1}$$

The STr denotes the super-trace which is a trace over spacetime and internal indices, including the appropriate multiplicities for non-scalar fields and a negative sign for Grassmann-valued fields. $\Gamma_k^{(2)}$ is the second functional derivative of $\Gamma_k$ with respect to the fields, i.e., it

---

[4]The existence of light fermions has also been studied in Loop Quantum Gravity [130], quadratic gravity [13] and dynamical triangulations as an approach to asymptotic safety [131].

is a matrix in field space with the individual entries transforming in the appropriate representations of spacetime and internal symmetry groups. The properties of the regulator function[5] ensure that the r.h.s. of Eq. (3.1) is both IR- and UV-finite and the contribution to $k\partial_k\Gamma_k$ is peaked for modes with (generalized) momenta of order $k^2$.

For practical purposes, we need to truncate $\Gamma_k$ in order to compute beta functions in a given physical system. The choice of truncation results in systematic uncertainties in the results. To keep systematic uncertainties under control, truncations must follow an ordering principle, such that terms beyond the truncation contribute to the results at a subleading order compared to term included in the truncation. Beyond strictly perturbative settings, finding such an ordering principle for truncations can be subtle. In gravity-matter systems, there is evidence for near-perturbative behavior. This is synonymous with scaling exponents exhibiting near-canonical values. Thus, canonically highly irrelevant operators are expected to stay irrelevant at an asymptotically safe gravity-matter fixed point, see, e.g., [56, 85, 133, 134] for evidence in pure-gravity systems and [19, 87] for evidence in gravity-matter systems. Further, near-perturbativity is supported by the result that symmetry identities for diffeomorphism symmetry remain close to trivial [100, 105, 106]. We follow this ordering principle in our choice of truncation and consider the leading order four-fermion interactions and a minimal coupling to gravity. Higher-order four-fermion interactions, e.g., including derivatives or curvature terms, are neglected.

In this work, we consider a system composed of (Dirac) fermions, Abelian (U(1)) and non-Abelian (SU($N_c$)) gauge fields, and gravitational degrees of freedom. The fermions are charged under U(1) and SU($N_c$), and we denote them as $\psi_{i,I}$ (with $i, j, \cdots \in \{1, \cdots, N_c\}$ denoting SU($N_c$)-indices in the fundamental representation and $I, J, \cdots \in \{1, \cdots, N_f\}$ denoting flavor indices). We use $A_\mu$ and $B_\mu^a$ (with $a, b, \cdots \in \{1, \cdots, N_c^2 - 1\}$) to denote components of Abelian and non-Abelian gauge fields, respectively.

In the gravitational sector, we use the Einstein-Hilbert truncation

$$\Gamma_k^{\mathrm{EH}} = \frac{k^2}{16\pi G_k} \int_x \sqrt{g}\left(2k^2\Lambda_k - R(g)\right),\tag{3.2}$$

where $G_k$ and $\Lambda_k$ denote the dimensionless counterpart of the scale-dependent Newton coupling and cosmological constant.

To define a local renormalization group flow in the presence of gravitational degrees of freedom, we need to introduce a background metric that allows us to define the notion of coarse-graining. Thus, we quantize gravity in terms of metric fluctuations $h_{\mu\nu}$ defined according to

$$g_{\mu\nu} = \delta_{\mu\nu} + Z_h^{1/2}(k^{-2}G_k)^{1/2}h_{\mu\nu},\tag{3.3}$$

where $\delta_{\mu\nu}$ stands for the flat (Euclidean) metric, $Z_h$ is the wave-function renormalization of $h_{\mu\nu}$ and we introduce the factor $(k^{-2}G_k)^{1/2}$ to fix the canonical mass dimension of $h_{\mu\nu}$ to one. We use the flat background because it is sufficient to extract the beta functions

---

[5]In general, we demand the following properties of $\mathbf{R}_k(p^2)$: i) for fixed $k^2$, it should decrease (monotonically) with $p^2$; ii) for fixed $p^2$, it should increase (monotonically) with $k^2$; iii) $\lim_{k\to 0} \mathbf{R}_k(p^2) = 0$ for all $p^2$; iv) for $p^2 > k^2$, $\mathbf{R}_k(p^2)$ should go to zero sufficiently fast.

of four-fermion and gauge couplings even in the presence of gravity. To compute the beta functions of the gravitational couplings $G_k$ and $\Lambda_k$, it is convenient to promote $\delta_{\mu\nu}$ to a curved background metric $\bar{g}_{\mu\nu}$.

In the fermionic sector, we use

$$
\begin{aligned}
\Gamma_k^{\text{ferm}} = \int_x \sqrt{g} \bigg( & Z_\psi \, i\bar{\psi}_{i,I}\gamma^\mu\nabla_\mu\psi_{i,I} + \frac{1}{2k^2}Z_\psi^2\lambda_{k,+}(\text{V}+\text{A}) + \frac{1}{2k^2}Z_\psi^2\lambda_{k,-}(\text{V}-\text{A}) \\
& + \frac{1}{2k^2N_\text{c}}Z_\psi^2\lambda_{k,\text{VA}}\Big((\text{V}-\text{A}) + 2N_\text{c}\,(\text{V}-A)_{\text{Adj.}}\Big) + \frac{1}{2k^2}Z_\psi^2\lambda_{k,\sigma}\big(\text{S}-\text{P}\big) \bigg),
\end{aligned} \tag{3.4}
$$

where $Z_\psi$ is the fermion wave-function renormalization, and $\lambda_{k,+}$, $\lambda_{k,-}$, $\lambda_{k,\text{VA}}$ and $\lambda_{k,\sigma}$ are scale-dependent dimensionless four-fermion couplings. The four-fermion interactions are

$$
\text{V} \pm \text{A} = \big(\bar{\psi}\gamma_\mu\psi\big)^2 \mp \big(\bar{\psi}\gamma_\mu\gamma_5\psi\big)^2 \tag{3.5a}
$$

$$
\text{V} - \text{A} + 2N_\text{c}\,(V-A)_{\text{Adj.}} = \big(\bar{\psi}T^a\gamma_\mu\psi\big)^2 + \big(\bar{\psi}T^a\gamma_\mu\gamma_5\psi\big)^2 \tag{3.5b}
$$

$$
\text{S} - \text{P} = \big(\bar{\psi}_{i,I}\psi_{i,J}\big)^2 - \big(\bar{\psi}_{i,I}\gamma_5\psi_{i,J}\big)^2 \tag{3.5c}
$$

If the color indices $(i, j...)$ and flavor indices $(I, J, ...)$ are not indicated explicitly in a bilinear, the bilinear is understood to be a color/flavor singlet. These four interactions form a Fierz-complete basis, i.e., any other four-fermion invariant without derivatives and curvature-dependence can be transformed into these four, see [135]. Working in a Fierz-complete basis is critical, because otherwise symmetry-breaking or -restoring effects can be missed.

Additionally, the covariant derivative $\nabla_\mu$ encodes the minimal coupling to gravity as well as to the Abelian and non-Abelian gauge fields, thus

$$
\nabla_\mu\psi_{i,I} = \partial_\mu\psi_{i,I} + ie\,A_\mu\psi_{i,I} - ig\,B_\mu^a T_{IJ}^a\psi_{i,J} + \omega_\mu\psi_{i,I}\,, \tag{3.6}
$$

where $e$ and $g$ are gauge couplings, $T_{IJ}^a$ denotes components of generators in the fundamental representation of $\text{SU}(N_\text{c})$ and $\omega_\mu$ is the spin-connection. We do not treat the spin-connection as an independent field. Instead, we express $\omega_\mu$ in terms of the Christoffel connection using the vierbein formalism, and write vierbein fluctuations in terms of metric fluctuations [136, 137].

In the gauge-field sector, we use the truncation

$$
\Gamma_k^{\text{gauge}} = \int_x \sqrt{g}\left(\frac{Z_A}{4}g^{\mu\alpha}g^{\nu\beta}F_{\mu\nu}F_{\alpha\beta} + \frac{Z_B}{4}g^{\mu\alpha}g^{\nu\beta}G_{\mu\nu}^a G_{\alpha\beta}^a\right), \tag{3.7}
$$

where $Z_A$ and $Z_B$ denote the wave-function renormalization of $A_\mu$ and $B_\mu^a$ respectively and $F_{\mu\nu}$ and $G_{\mu\nu}^a$ denote the field-strength tensors associated with $A_\mu$ and $B_\mu^a$.

The full truncation for $\Gamma_k$ reads

$$
\Gamma_k = \Gamma_k^{\text{EH}} + \Gamma_k^{\text{ferm}} + \Gamma_k^{\text{gauge}} + \Gamma_k^{\text{g.f.}}\,, \tag{3.8}
$$

where we also added gauge-fixing terms for the gravitational and (non-)Abelian gauge sectors, namely

$$
\begin{aligned}
\Gamma_k^{\text{g.f.}} = \frac{Z_h}{2\alpha_h}\int_x & \left(\partial^\alpha h_{\mu\alpha} - \frac{1+\beta_h}{4}\partial_\mu h^\alpha{}_\alpha\right)\left(\partial_\beta h^{\nu\beta} - \frac{1+\beta_h}{4}\partial^\nu h^\beta{}_\beta\right) \\
& + \frac{Z_A}{2\alpha_A}\int_x(\partial_\mu A^\mu)^2 + \frac{Z_B}{2\alpha_B}\int_x(\partial_\mu B^{\mu a})^2 + \Gamma_k^{\text{ghost}}\,.
\end{aligned} \tag{3.9}
$$

Herein $\alpha_h$ $\alpha_A$, $\alpha_B$ and $\beta_h$ denote gauge-fixing parameters. Throughout this paper we consider the gauge choice $\alpha_{h,A,B} \to 0$ and $\beta_h \to 0$. We also add a *Mathematica* notebook as a supplementary material containing beta functions with full $\beta_h$ dependence. In addition, $\Gamma_k^{\text{ghost}}$ represents the corresponding Faddeev-Popov action, but this sector is not relevant for the results presented in this paper.

In explicit calculations with the FRG, one needs to specify the form of the regulator function $\mathbf{R}_k(p^2)$. Throughout this paper we work with the following regulator:

$$\mathbf{R}_k(p^2) = \left( \Gamma_{k,0}^{(2)}(p^2) - \Gamma_{k,0}^{(2)}(0) \right) r(p^2/k^2), \tag{3.10}$$

where the subscript "0" in $\Gamma_{k,0}^{(2)}$ indicates that we are evaluating the 2-point function at vanishing field configurations. We use the Litim shape function [138]

$$r(y) = \left( \frac{1}{y} - 1 \right) \theta(1 - y), \tag{3.11}$$

except for spinor fields, where we use the shape function (*ibid.*)

$$r_\psi(y) = \left( \frac{1}{\sqrt{y}} - 1 \right) \theta(1 - y). \tag{3.12}$$

## 4 Beta functions

We have computed the beta function of the four-fermions couplings $\lambda_+$, $\lambda_-$, $\lambda_{\text{VA}}$ and $\lambda_\sigma$ including contributions from gravity and gauge fields $A_\mu$ and $B_\mu^a$. In the following, we present the explicit expressions, where we neglect anomalous dimension contributions coming from the insertion of $k\partial_k \mathbf{R}_k$. This is justified as long as the anomalous dimension stays sufficiently small

$$
\begin{aligned}
\beta_{\lambda_+} = {} & (2 + \eta_{4\text{ferm}}) \lambda_+ + \frac{3\lambda_+^2}{8\pi^2} + \frac{(1 + N_c N_f) \lambda_+ \lambda_-}{4\pi^2} + \frac{(N_c + N_f) \lambda_+ \lambda_{\text{VA}}}{4\pi^2} \\
& - \frac{N_f \lambda_- \lambda_\sigma}{8\pi^2} - \frac{\lambda_{\text{VA}} \lambda_\sigma}{8\pi^2} - \frac{\lambda_\sigma^2}{8\pi^2} + \frac{3 e^2 \lambda_+}{4\pi^2} + \frac{3 g^2 \lambda_+}{8\pi^2 N_c} + \frac{9 e^4}{32\pi^2} + \frac{9 (4 + N_c^2) g^4}{512\pi^2 N_c^2} - \frac{9 e^2 g^2}{32\pi^2 N_c} \\
& + \frac{5 G^2}{8(1 - 2\Lambda)^3} + \frac{5 e^2 G}{16\pi(1 - 2\Lambda)} + \frac{5 e^2 G}{16\pi(1 - 2\Lambda)^2} - \frac{3 e^2 G}{160\pi(1 - 4\Lambda/3)^2} + \frac{3 e^2 G}{160\pi(1 - 4\Lambda/3)} \\
& - \frac{5 g^2 G}{32\pi(1 - 2\Lambda)N_c} - \frac{3 g^2 G}{320\pi N_c (1 - 4\Lambda/3)} - \frac{5 g^2 G}{32\pi N_c(1 - 2\Lambda)^2} + \frac{3 g^2 G}{320\pi N_c (1 - 4\Lambda/3)^2},
\end{aligned} \tag{4.1}
$$

$$
\begin{aligned}
\beta_{\lambda_-} = {} & (2 + \eta_{4\text{ferm}}) \lambda_- - \frac{(1 - N_c N_f) \lambda_-^2}{8\pi^2} + \frac{N_c N_f \lambda_+^2}{8\pi^2} + \frac{N_c \lambda_- \lambda_{\text{VA}}}{4\pi^2} - \frac{N_f \lambda_+ \lambda_\sigma}{8\pi^2} \\
& + \frac{N_f \lambda_- \lambda_{\text{VA}}}{4\pi^2} - \frac{\lambda_{\text{VA}}^2}{4\pi^2} + \frac{3e^2 \lambda_-}{4\pi^2} - \frac{3 g^2 \lambda_-}{8\pi^2 N_c} + \frac{3 g^2 \lambda_{\text{VA}}}{8\pi^2} - \frac{9 (3N_c^2 + 4) g^4}{512\pi^2 N_c^2} - \frac{9 e^4}{32\pi^2} + \frac{9 e^2 g^2}{32\pi^2 N_c} \\
& - \frac{5 G^2}{8(1 - 2\Lambda)^3} + \frac{5 e^2 G}{16\pi(1 - 2\Lambda)} + \frac{3 e^2 G}{160\pi(1 - 4\Lambda/3)} + \frac{5 e^2 G}{16\pi(1 - 2\Lambda)^2} - \frac{3 e^2 G}{160\pi(1 - 4\Lambda/3)^2} \\
& - \frac{5 g^2 G}{32\pi N_c(1 - 2\Lambda)} - \frac{3 g^2 G}{320\pi N_c(14\Lambda/3)} - \frac{5 g^2 G}{32\pi N_c(1 - 2\Lambda)^2} + \frac{3 g^2 G}{320\pi N_c(1 - 4\Lambda/3)^2},
\end{aligned} \tag{4.2}
$$

$$\beta_{\lambda_\sigma} = (2 + \eta_{4\text{ferm}}) \lambda_\sigma - \frac{N_{\mathrm{c}} \lambda_\sigma^2}{4\pi^2} + \frac{N_{\mathrm{f}} \lambda_\sigma \lambda_{\mathrm{VA}}}{4\pi^2} + \frac{\lambda_- \lambda_\sigma}{4\pi^2} + \frac{3 \lambda_+ \lambda_\sigma}{4\pi^2}$$
$$- \frac{3 e^2 \lambda_\sigma}{4\pi^2} + \frac{3 g^2 (1 - N_{\mathrm{c}}^2) \lambda_\sigma}{8\pi^2 N_{\mathrm{c}}} + \frac{3 g^2 \lambda_+}{4\pi^2} + \frac{9 g^4 (8 - 3N_{\mathrm{c}}^2)}{256\pi^2 N_{\mathrm{c}}} - \frac{9 e^2 g^2}{16\pi^2} \tag{4.3}$$
$$- \frac{5 g^2 G}{16\pi(1 - 2\Lambda)} - \frac{3 g^2 G}{160\pi(1 - 4\Lambda/3)} - \frac{5 g^2 G}{16\pi(1 - 2\Lambda)^2} + \frac{3 g^2 G}{160\pi(1 - 4\Lambda/3)^2} \, ,$$

$$\beta_{\lambda_{\mathrm{VA}}} = (2 + \eta_{4\text{ferm}}) \lambda_{\mathrm{VA}} + \frac{(N_{\mathrm{c}} + N_{\mathrm{f}}) \lambda_{\mathrm{VA}}^2}{8\pi^2} + \frac{N_{\mathrm{f}} \lambda_\sigma^2}{32\pi^2} - \frac{\lambda_- \lambda_{\mathrm{VA}}}{2\pi^2}$$
$$+ \frac{3 e^2 \lambda_{\mathrm{VA}}}{4\pi^2} - \frac{3 g^2 \lambda_{\mathrm{VA}}}{8\pi^2 N_{\mathrm{c}}} + \frac{3 g^2 \lambda_-}{8\pi^2} - \frac{9 g^4 N_{\mathrm{c}}}{512\pi^2} + \frac{9 g^4}{64\pi^2 N_{\mathrm{c}}} - \frac{9 e^2 g^2}{32\pi^2} \tag{4.4}$$
$$+ \frac{5 g^2 G}{32\pi(1 - 2\Lambda)} + \frac{3 g^2 G}{320\pi(1 - 4\Lambda/3)} + \frac{5 g^2 G}{32\pi(1 - 2\Lambda)^2} - \frac{3 g^2 G}{320\pi(1 - 4\Lambda/3)^2} \, .$$

Because gravity couples to the spacetime indices, not the internal indices of matter fields, it is "blind" to internal symmetries. Therefore, the gravitational contribution to the scaling dimension at a free fixed point of the four different 4-fermion channels is the same, i.e., there is a Fierz-universality of the gravitational contribution, see [14] for more details. This contribution is

$$\eta_{4\text{ferm}} = 2 \eta_\psi + \frac{5 G}{2\pi(1 - 2\Lambda)^2} - \frac{G}{20\pi(1 - 4\Lambda/3)} - \frac{31 G}{60\pi(1 - 4\Lambda/3)^2} \, . \tag{4.5}$$

The one-loop fermion anomalous $\eta_\psi$ receives contributions only from gravity, resulting in

$$\eta_\psi = \frac{3 G}{20\pi(1 - 4\Lambda/3)} - \frac{25 G}{16\pi(1 - 2\Lambda)^2} + \frac{29 G}{80\pi(1 - 4\Lambda/3)^2} \, . \tag{4.6}$$

The gravitational contribution to the four-fermion couplings $\lambda_+$ and $\lambda_-$ was first computed in [8], where the anomalous dimension of the fermion was not computed, but instead treated as a free parameter. In [10], the anomalous dimension was added and in [11], the contribution from an Abelian gauge field together with gravity was accounted for.

To show the effect of the fermion anomalous dimension, we evaluate the anomalous dimension of the four-fermion couplings at $\lambda_i = 0$ (i.e., $\eta_{4\text{ferm}}$), with and without the anomalous dimension $\eta_\psi$, cf. Fig. 1.

In part of our analysis we explicitly use the fixed-point values for the gravitational couplings $G$ and $\Lambda$ as a function of the number of fermions and gauge fields. To compute the fixed points of the gravitational couplings, we use the beta functions

$$\beta_G = 2 G - \frac{5 G^2}{6\pi(1 - 2\Lambda)} - \frac{5 G^2}{3\pi(1 - 2\Lambda)^2} + \frac{G^2}{6\pi(1 - 4\Lambda/3)}$$
$$- \frac{(11 + 32 \log(3/2)) G^2}{12\pi} + \frac{(2N_{\mathrm{D}} - 4N_{\mathrm{V}}) G^2}{6\pi} \, , \tag{4.7a}$$

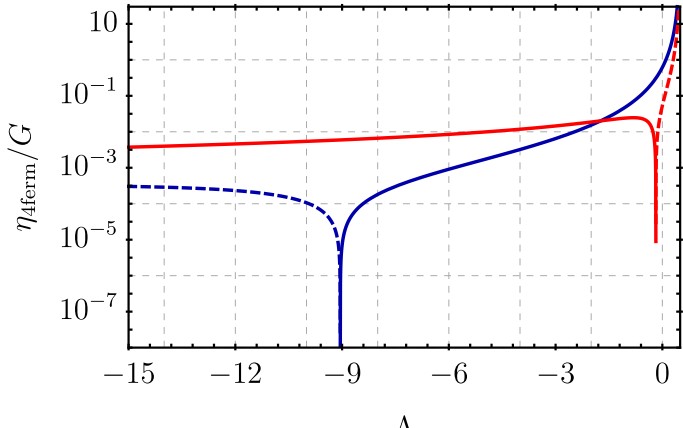

**Figure 1**. We show the gravitational contribution to the anomalous dimension of the four-fermion couplings,*i.e.*, $\eta_{\text{4ferm}}$, as a function of the cosmological constant $\Lambda$. The red line correspond to $\eta_{\text{4ferm}}$ including the fermion anomalous dimension $\eta_{\psi}$, as in [10, 11]. The blue line is $\eta_{\text{4ferm}}$ neglecting $\eta_{\psi}$-terms, as in [8]. The dashed lines correspond to $-\eta_{\text{4ferm}}$.

$$
\begin{aligned}
\beta_{\Lambda} = {} & -2\Lambda + \frac{5\,G}{4\pi(1-2\Lambda)} - \frac{5\,\Lambda\,G}{6\pi(1-2\Lambda)} - \frac{5\,\Lambda\,G}{3\pi(1-2\Lambda)^2} \\
& + \frac{G}{4\pi\,(1-4\Lambda/3)} + \frac{\Lambda\,G}{6\pi\,(1-4\Lambda/3)} - \frac{7G}{2\pi} + \frac{4\log\,(3/2)\,G}{\pi} \\
& - \frac{(11+32\log\,(3/2))\,\Lambda\,G}{12\pi} - \frac{(4\,N_{\text{D}}-2\,N_{\text{V}})\,G}{4\pi} + \frac{(2\,N_{\text{D}}-4\,N_{\text{V}})\,\Lambda\,G}{6\pi} ,
\end{aligned}
\tag{4.7b}
$$

where $N_{\text{D}} = N_{\text{f}} \cdot N_{\text{c}}$ is the number of Dirac fermions and $N_{\text{V}} = 1 + (N_{\text{c}}^2 - 1)$ is the number of vector fields. The pure-gravity and fermionic parts were taken from [12], while the vector contributions come from [89].

## 5 Fixed-point collisions: Complex action, stability trading or unchanged universality class

In the literature, the collision of fixed points is well explored, because it underlies several important mechanisms, both in statistical physics and high-energy physics, including, e.g., the exchange of universality classes in statistical physics [139–142] and the weak-gravity bound in gravity [10, 29, 31, 33, 103, 143].

After such a collision, one of three scenarios is realized, cf. Fig. 2:

(i) The two fixed points have become complex. Following the RG flow towards the IR, the corresponding coupling thus increases in absolute value without bound (towards positive values, if the beta function is negative, and towards negative values, if the beta function is positive).

(ii) The two fixed points remain real and have exchanged their stability properties. This second scenario can be realized in two distinct ways:

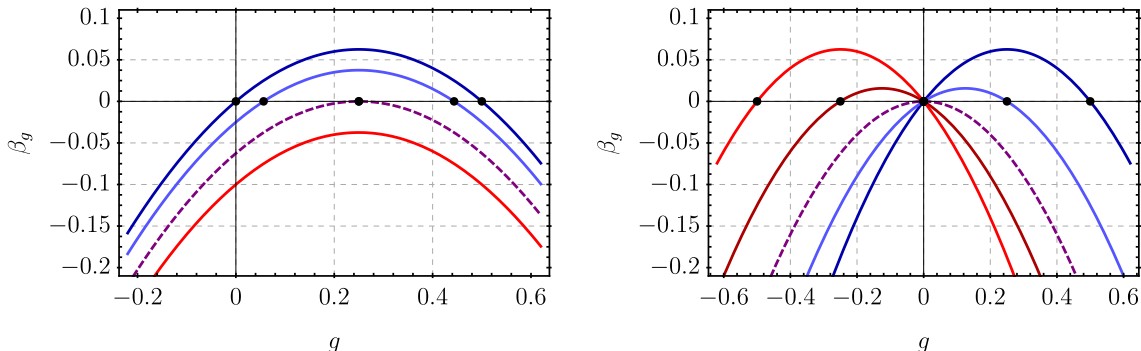

**Figure 2**. We illustrate scenarios (i) and (ii) for fixed-point collisions, using a simple schematic beta function of the form $\beta_g = a + b\,g + c\,g^2$. We set $c = -1$ in both cases.
Left panel: scenario (i) is realized by varying $a$ between $a = 0$ (blue line) and $a = -1/10$ (red line) at $b = 1/2$, with the fixed-point collision occurring at $a = -1/16$ (purple dashed line).
Right panel: scenario (ii) is realized by varying $b$ between $b = 1/2$ (blue line) and $b = -1/2$ (red line) at $a = 0$, with the fixed-point collision occurring at $b = 0$ (purple dashed line).

(a) If the two fixed points are characterized by different symmetry properties, the two universality classes can be distinguished. Such a fixed-point collision generically changes the phenomenology of the system, e.g., systems in statistical physics then undergo phase transitions with distinct properties [141, 142].

(b) If the two fixed points are characterized by the same symmetry properties, i.e., lie in the same theory space, they cannot be distinguished. If the minimal number of relevant directions of any of the two fixed points is preserved under the collision, the universality class is unchanged.

Which of the cases is realized, depends on the coefficients in the beta function and how they change as a function of the external parameter. As the simple examples in Fig. 2 illustrates, scenario (i) is driven by changes in the term that generates the interaction (the zeroth-order term in the beta function), whereas scenario (ii) is driven by changes in the scaling dimension of the coupling (by which we mean the linear term in the coupling's beta function). Generically, gravity contributes to both terms and which scenario is realized therefore depends on both the signs of these terms as well as their relative size.

Different criteria can be used to find parameter values at which fixed-point collisions occur, depending on the scenario.

Fixed-point collisions are always linked to at least one critical exponent[6] being zero. This is simplest to see for a single beta function, at which a fixed-point collision is linked to a degenerate zero and thus a vanishing critical exponent. When there is more than one coupling in the system, there is one direction in the space of couplings along which the two fixed points approach each other; typically this direction is not aligned with any of the couplings. This direction corresponds to an eigendirection of the fixed point at the moment of collision; thus, there is again a vanishing critical exponent. One can therefore identify

---

[6]Our conventions are such that a *relevant* coupling has *positive* critical exponent.

the parameter values at which the collision occurs for all three scenarios by searching for zeros of the eigenvalues of a fixed point.

To distinguish which of the three scenarios is realized, one must go beyond the analysis of the critical exponents. Scenario (i) is uniquely identified by the imaginary part of the fixed-point values becoming non-zero. This criterion, i.e., searching for parameter values at which the fixed point acquires a non-zero imaginary part, will not detect the fixed-point collision in scenarios (ii); instead, (ii) can be detected by searching for parameter values where fixed-point coordinates are equal for two different fixed points and/or by investigating the critical exponents. To distinguish (ii a) from (ii b), one must finally investigate which fixed point has the lowest number of relevant directions before and after the collision. By comparing the symmetry properties of the fixed-point action one can decide which scenario within (ii) is realized.

## 6 Fixed-point collision without chiral symmetry breaking for uncolored, uncharged fermions

Chiral symmetry breaking is tied to fixed-point collisions: four-fermion couplings are driven to criticality, when the interactions become relevant. To become relevant, the sign of their scaling dimension has to change. In this process (controlled by, e.g., the value of the gauge coupling treated as an external parameter), the scaling dimension becomes zero. In turn, a vanishing scaling dimension implies a degenerate zero of the beta function, i.e., a collision of two fixed points.[7] This collision occurs when the interaction that drives the system to criticality (e.g., a gauge interaction) is treated as an external parameter.

Of the three scenarios for fixed-point collisions, in gravity-matter systems, (i) has been discovered in several settings, giving rise to a weak-gravity bound that limits the values of the gravitational couplings so that a fixed point persists within the studied setting.[8] Purely fermionic systems coupled to gravity appear to be an exception, with no indications for scenario (i) discovered to date. Instead, in [8] it was already found that for specific values of the fermion anomalous dimension, (ii b) may be realized.

Here, we strengthen the evidence that uncolored, uncharged fermions are protected from chiral symmetry breaking, despite undergoing a gravity-induced fixed-point collision, cf. Fig. 3. This is linked to how gravity enters a beta function, where it generates a shift in fixed-point values, but also changes the scaling dimension, such that the fixed-point values remain real.

If one demands continuous differentiability of the fixed-point solution as a function of $G$, one can single out which of the two solutions corresponds to the shifted Gaussian fixed point[9] after the collision; it is the one with one relevant direction. We are, however,

---

[7]There is an exception, namely the case of a complex pair of critical exponents. The real part can transition to zero, if the imaginary part is nonzero, without implying a fixed-point collision.

[8]In [103], it has been found that the fixed-point collision does not persist at higher truncation orders because the system actually only features a single fixed point; instead, the real part of the critical exponent changes sign which enables a switch to relevance without a fixed-point collision.

[9]The shifted Gaussian fixed point is the fixed point that becomes Gaussian when gravitational interactions are switched off, i.e., in the limit $G \to 0$.

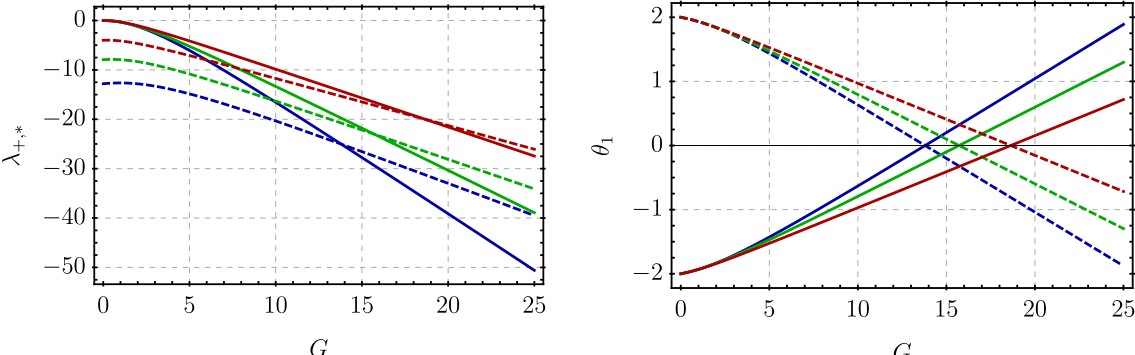

**Figure 3**. Left panel: We show the fixed-point value of $\lambda_+$ at the shifted Gaussian fixed point (continuous lines) and one of the non-Gaussian fixed points (dashed lines), for $N_{\rm f} = 6$ (dark blue lines), $N_{\rm f} = 10$ (green lines) and $N_{\rm f} = 20$ (red lines). The two fixed points undergo a collision, and both remain real. Beyond the collision, we identify which fixed point to call the shifted Gaussian, and which one the non-Gaussian, by demanding continuous differentiability of the fixed-point solution as a function of $G$.
Right panel: We show the critical exponent that is zero at the fixed-point collision. The line-styles are the same as for the left panel.

not aware that imposing continuous differentiability as a function of an external parameter is justified. For the physical consequences of asymptotic safety, the only relevant piece of information is that at all values of $G$, there is a fixed point with only irrelevant directions.

As we will discuss below, scenario (ii b) changes to scenario (ii a), as soon as the fermionic system has enough structure to allow for distinct symmetry-properties of fixed points.

## 7 Global color charge and light fermions

At a first glance, one might expect that color is only a spectator in the gravity-fermion system and does not change the results. This, however, is not the case, because of the additional four-fermion interactions which are Fierz-independent in the presence of color. The theory space of colored chiral fermions contains four distinct interactions, see [7]:

$$\lambda_\pm \left( \left( \bar{\psi} \gamma_\mu \psi \right)^2 \mp \left( \bar{\psi} \gamma_\mu \gamma_5 \psi \right)^2 \right), \tag{7.1}$$

$$\lambda_\sigma \left( \left( \bar{\psi}_{i,I} \psi_{i,J} \right)^2 + \left( \bar{\psi}_{i,I} \gamma_5 \psi_{i,J} \right)^2 \right), \tag{7.2}$$

$$\lambda_{\rm VA} \left( \left( \bar{\psi} T^a \gamma_\mu \psi \right)^2 + \left( \bar{\psi} T^a \gamma_\mu \gamma_5 \psi \right)^2 \right). \tag{7.3}$$

This theory space has several symmetry-enhanced subspaces, see Tab. 1. First, if $\lambda_\sigma$ and $\lambda_{\rm VA}$ are both set to zero, color and flavor indices can be combined into one "superindex" $\alpha = 1, \ldots N_{\rm c} \cdot N_{\rm f}$ and there is a $U(N_{\rm f} N_{\rm c})_{\rm L} \times U(N_{\rm f} N_{\rm c})_{\rm R}$ symmetry that rotates all $N_{\rm c} \cdot N_{\rm f}$ components of the left- as well as the right-handed fermion separately. In the presence of non-zero $\lambda_{\rm VA}$, the $N_{\rm f}$ left- and right-handed flavors can be rotated into each other separately

| Subspace | Symmetry |
|----------|----------|
| $\lambda_{\mathrm{VA}} = \lambda_\sigma = 0$ | $\mathrm{U}(N_\mathrm{f} N_\mathrm{c})_\mathrm{L} \times \mathrm{U}(N_\mathrm{f} N_\mathrm{c})_\mathrm{R}$ |
| $\lambda_{\mathrm{VA}} \neq 0, \lambda_\sigma = 0$ | $\mathrm{U}(N_\mathrm{f})_\mathrm{L} \times \mathrm{U}(N_\mathrm{f})_\mathrm{R} \times (\mathrm{U}(N_\mathrm{c})_\mathrm{L} \times \mathrm{U}(N_\mathrm{c})_\mathrm{R})^{N_\mathrm{f}}$ |
| $\lambda_{\mathrm{VA}} \neq 0, \; \lambda_\sigma \neq 0$ | $\mathrm{U}(N_\mathrm{f})_\mathrm{L} \times \mathrm{U}(N_\mathrm{f})_\mathrm{R} \times \mathrm{U}(N_\mathrm{c})$ |

**Table 1**. Subspaces with enhanced global symmetries in the absence of the non-abelian gauge coupling (i.e., $g = 0$).

and there is an additional global color rotation for each flavor and each chirality. Finally, in the presence of all four channels, there are the left- and right-handed flavor rotations and there is the global $U(N_\mathrm{c})$. In the presence of nonzero non-Abelian gauge coupling, the $\mathrm{SU}(N_\mathrm{c})$ subgroup becomes local.[10]

Gravity operates differently in these subspaces: it only generates interactions in the most symmetric subspace because the largest symmetry, $\mathrm{U}(N_\mathrm{f} N_\mathrm{c})_\mathrm{L} \times \mathrm{U}(N_\mathrm{f} N_\mathrm{c})_\mathrm{R}$, is always the symmetry of the kinetic term. There is therefore no Gaussian fixed point in the presence of gravity, because the Gaussian fixed point gets shifted to nonzero $\lambda_\pm$.

For the remaining four-fermion interactions, gravity only affects the scaling dimension, but allows the couplings to be zero. Thus, a subset of fixed points lives in the $\lambda_\sigma = \lambda_{\mathrm{VA}} = 0$ theory space.

There are additional fixed points outside this subspace which have nonzero couplings already in the absence of gravity. Their positions and scaling exponents are affected by gravity as well and therefore they can play an important role. They take part in fixed-point collisions with the fixed points in the most symmetric subspace. In doing so, they change the critical exponent of the symmetry-enhanced interactions from negative to positive. As a consequence, the symmetry of the system that is realized at low energies changes.

We focus on the case $N_\mathrm{f} = 6$ and $N_\mathrm{c} = 3$ in the following, taking inspiration from the six flavors and three colors of quarks in the SM. The shifted Gaussian fixed point undergoes a collision with six other fixed points, cf. Fig. 4 at a critical value of $G = G_{\mathrm{crit}}$. At the collision, three of its critical exponents acquire a positive real part, cf. Fig. 5. Thus, in particular, one of the critical exponents within the symmetry-enhanced $(\lambda_+, \lambda_-)$ plane also changes sign.

For $G < G_{\mathrm{crit}}$ there is an enhanced symmetry that is emergent under the RG flow to the IR, because both $\lambda_\sigma$ and $\lambda_{\mathrm{VA}}$ are driven to zero by the RG flow, cf. Fig. 6. This symmetry-enhancement already occurs in a system without gravity, due to the canonical dimension of $\lambda_\sigma$ and $\lambda_{\mathrm{VA}}$. Under the impact of quantum gravity, the approach to the symmetric hyperspace is decelerated, because gravity pushes the critical exponents closer to zero.

Beyond $G = G_{\mathrm{crit}}$, a fixed point with four irrelevant directions remains which generically attracts RG trajectories. Accordingly, four-fermion interactions are still protected from diverging, even at $G > G_{\mathrm{crit}}$. The fixed point with four irrelevant directions still fea-

---

[10] Recall that gauge fields correspond to Lie algebras, not groups. For each connected component of a Lie group, one needs to introduce a new gauge group. The fact that it is the $\mathrm{SU}(N_\mathrm{c})$ subgroup that gets gauged while the $\mathrm{U}(1)$ subfactor remains ungauged can be fixed pragmatically by appealing to experiment.

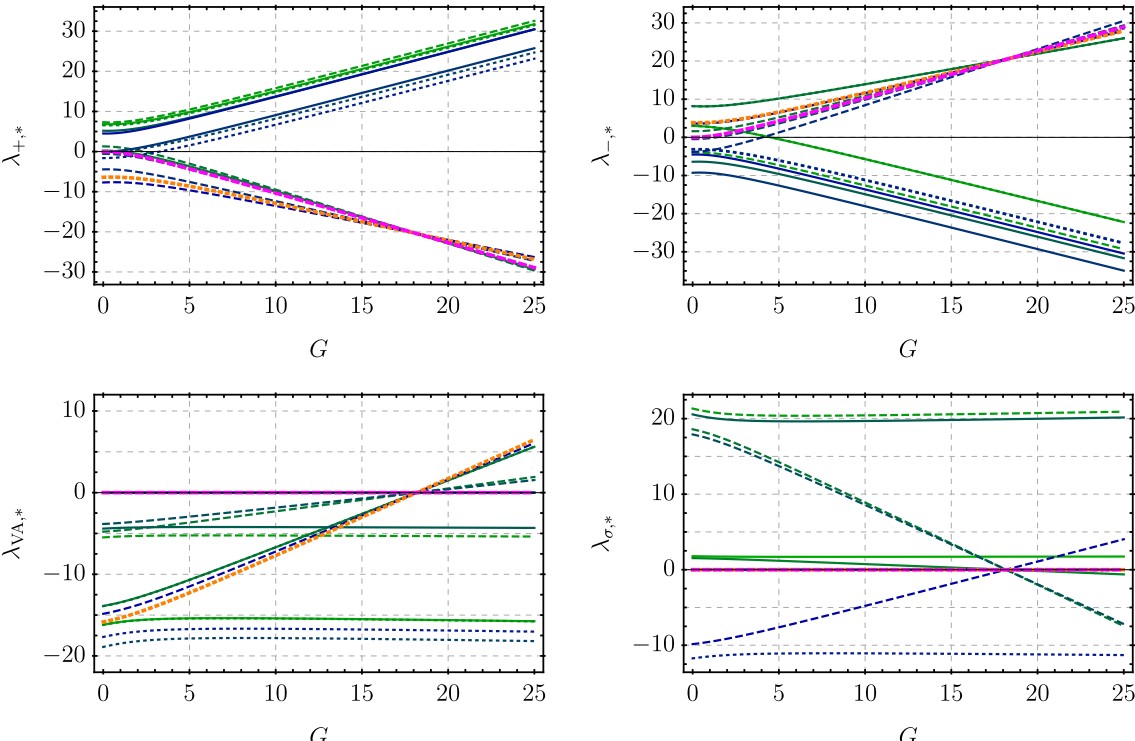

**Figure 4**. We show the fixed-point values for all four-fermion couplings as a functional of the Newton coupling $G$ (with $\Lambda = 0$). Different lines correspond to different fixed points. The shifted Gaussian fixed point is the magenta (dashed) line. We note that the shifted Gaussian fixed point has $\lambda_\sigma = \lambda_{\mathrm{VA}}$ for all values of $G$, which is consistent with the maximal symmetry $\mathrm{U}(N_\mathrm{f}N_\mathrm{c})_\mathrm{L} \times \mathrm{U}(N_\mathrm{f}N_\mathrm{c})_\mathrm{R}$. We also highlight, with orange (dotted) line, the non-Gaussian fixed point that exchange stability properties with the shifted Gaussian one after the fixed-point collision, c.f. Eq. (5). This non-Gaussian fixed point has $\lambda_\sigma = 0$ and, therefore, it belongs to a theory space symmetric under $\mathrm{U}(N_\mathrm{f})_\mathrm{L} \times \mathrm{U}(N_\mathrm{f})_\mathrm{R} \times (\mathrm{U}(N_\mathrm{c})_\mathrm{L} \times \mathrm{U}(N_\mathrm{c})_\mathrm{R})^{N_\mathrm{f}}$, c.f. Tab. 1.

tures symmetry enhancement, albeit of a lesser degree, namely to a $\mathrm{U}(N_\mathrm{f})_\mathrm{L} \times \mathrm{U}(N_\mathrm{f})_\mathrm{R} \times (\mathrm{U}(N_\mathrm{c})_\mathrm{L} \times \mathrm{U}(N_\mathrm{c})_\mathrm{R})^{N_\mathrm{f}}$ global symmetry. This is the case because $\lambda_{\sigma,*} = 0$ at the fixed point. In turn, the result $\lambda_{\sigma,*} = 0$ can already be expected from the beta function Eq. (4.3): for the case of global color symmetry (i.e., $g = 0$), $\lambda_{\sigma *} = 0$ is clearly always available as a fixed point. It is IR attractive, as long as the canonical scaling term $2\lambda_\sigma$ is not overwhelmed by terms encoding fluctuations and of the opposite sign. However, all terms encoding fluctuations are schematically of the form $\frac{\lambda_\mathrm{i}\lambda_\sigma}{4\pi^2}$; thus, as long as the fixed-point values for the other four-fermion couplings remain smaller than $4\pi^2$, the canonical scaling term dominates. This explains the observed result, namely that $\lambda_{\sigma,*} = 0$ remains IR attractive and the symmetry is always enhanced, even for $G > G_\mathrm{crit}$.

To summarize, there is an interplay between the global color symmetry and gravity: If gravity is weakly coupled (in the sense of $G < G_\mathrm{crit}$), the largest possible global symmetry that combines color and flavor emerges in the IR. If gravity is strongly coupled (in the sense of $G > G_\mathrm{crit}$), a smaller global symmetry which does not mix color with flavor, emerges in

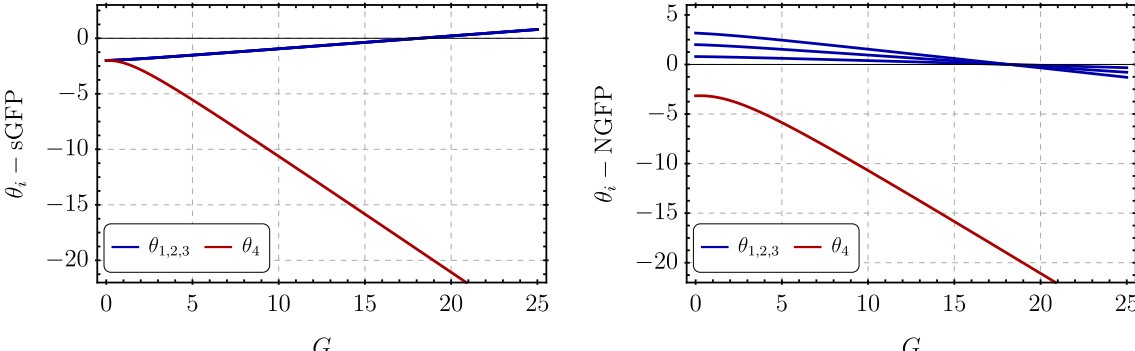

**Figure 5**. Left panel: We show the critical exponents associated with the shifted Gaussian fixed point as a function of $G$. Right panel: We show the critical exponents of the non-Gaussian fixed point highlighted in orange (dotted) in Fig. 4 as a function of $G$.

In both panels, we see that three of the critical exponents flip their signs at the same value of $G$, which is the same value of $G$ corresponding to the fixed-point collision shown in Fig. 4.

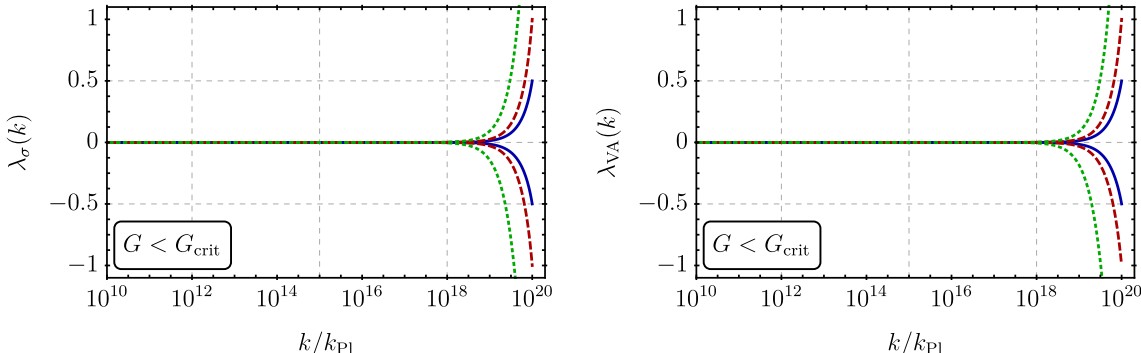

**Figure 6**. We show examples of RG trajectories with initial conditions in the UV lying outside the theory space with symmetry enhancement, i.e., we use $\lambda_\sigma(k_{UV}) \neq 0$ and $\lambda_{VA}(k_{UV}) \neq 0$ as initial conditions. The different lines correspond to different values of $\lambda_\sigma(k_{UV})$ and $\lambda_{VA}(k_{UV})$. In all cases, we set $G = 5$, which is smaller than $G_{crit}$ defined by the fixed-point collision.

We see that $U(N_f)_L \times U(N_f)_R \times U(N_c) \mapsto U(N_f N_c)_L \times U(N_f N_c)_R$ symmetry emerges, as $\lambda_\sigma$ and $\lambda_{VA}$ go to zero in the IR. The emergent symmetries are strongly IR attractive, such that $\lambda_\sigma$ and $\lambda_{VA}$ are driven to zero very quickly.

the IR. The enhanced symmetry contains a separate color rotation for each flavor.

Thus, the case of fermions charged under a global $SU(N_c)$ symmetry coupled to gravity realizes scenario (ii a) in the list of scenarios for fixed-point collisions (cf. Sec. 5).

## 8   Local color, U(1) charge and light fermions

The combined effect of gravitational, Abelian and non-Abelian gauge field fluctuations can trigger chiral symmetry breaking. The fully IR attractive shifted Gaussian fixed point only exists at real values in coupling space, when both gauge couplings and the gravitational

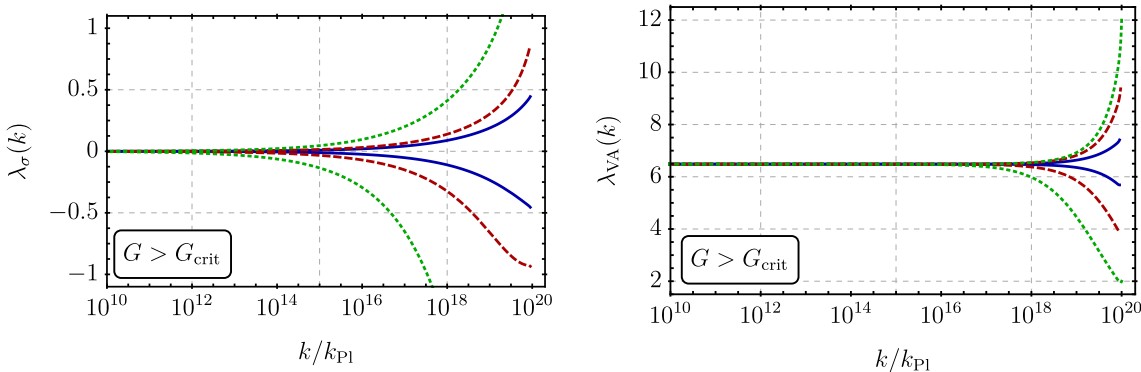

**Figure 7**. We show examples of RG trajectories with initial conditions $\lambda_\sigma(k_{\rm UV}) \neq 0$ and $\lambda_{\rm VA}(k_{\rm UV}) \neq 0$. The different lines correspond to different values of $\lambda_\sigma(k_{\rm UV})$ and $\lambda_{\rm VA}(k_{\rm UV})$. In all cases, we set $G = 25$, which is larger than $G_{\rm crit}$ defined by the fixed-point collision.

In contrast with the result show in Fig. 6, in the right panel we see that $\lambda_{\rm VA}$ tends to a non-vanishing value in the IR. Thus, this result shows that the RG flow leads to the symmetry enhancement $\mathrm{U}(N_{\rm f})_{\rm L} \times \mathrm{U}(N_{\rm f})_{\rm R} \times \mathrm{U}(N_{\rm c}) \mapsto \mathrm{U}(N_{\rm f})_{\rm L} \times \mathrm{U}(N_{\rm f})_{\rm R} \times (\mathrm{U}(N_{\rm c})_{\rm L} \times \mathrm{U}(N_{\rm c})_{\rm R})^{N_{\rm f}}$, which is weaker than the symmetry enhancement observed with $G < G_{\rm crit}$. This is a consequence of the exchange of stability properties between the shifted Gaussian fixed point and one of the non-Gaussian fixed points after the collision at $G = G_{\rm crit}$ (c.f. Figs. 4 and 5).

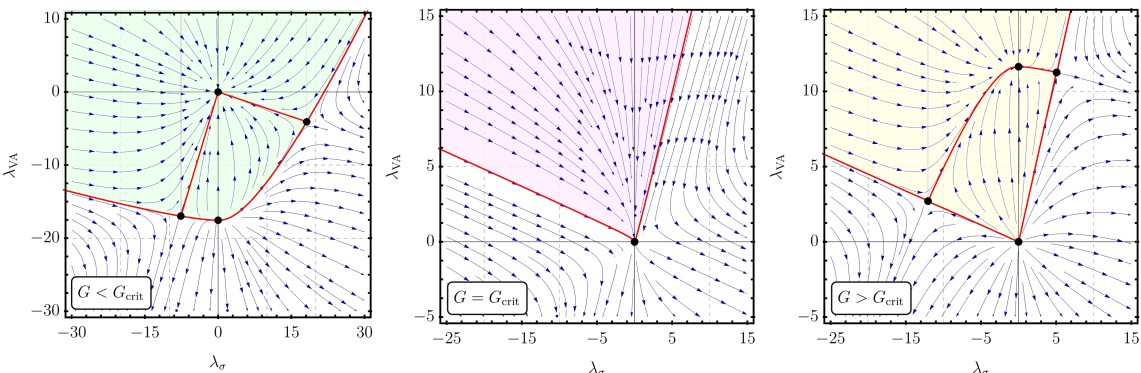

**Figure 8**. Phase diagrams obtained with the two-channel approximation where we consider only the flow of $\lambda_\sigma$ and $\lambda_{\rm VA}$. Despite the approximation, this reduced system captures the important qualitative features about the symmetry enhancement produced by the RG flow, which is also present in the full system

Left panel: we show the phase diagram obtained with $G = 0 < G_{\rm crit}$. The region in green represented the basin of attraction of the IR fixed point with $\lambda_\sigma = \lambda_{\rm VA} = 0$. Center panel: we show the phase diagram obtained with $G = 12\pi = G_{\rm crit}$. In this case, all the fixed point collide and the phase diagram exhibits a degenerate fixed point with $\lambda_\sigma = \lambda_{\rm VA} = 0$. The region in purple indicates the basin of attraction of such a fixed point. Right panel: we show the phase diagram obtained with $G \approx 62.7 > G_{\rm crit}$, thus, after the fixed point collision. In this case, the fixed point with $\lambda_\sigma = 0$ and $\lambda_{\rm VA} > 0$ becomes the most IR-attractive. The yellow regions correspond to its basin of attraction.

coupling are sufficiently small. Strong gravitational and/or non-Abelian fluctuations shift

the fixed point into the complex plane. Thus, in the presence of a local non-Abelian gauge symmetry, scenario (i) from the list of possible scenarios for fixed-point collisions is realized (cf. Sec. 5).

Since the color gauge degree of freedom is the main new ingredient, we first consider solely the effect of non-Abelian gauge and gravity fluctuations in Sec. 8.1, setting the Abelian gauge coupling $e = 0$. We convert these insights into upper bounds for the Planck-scale values of the non-Abelian gauge couplings in Sec. 8.2, without and with Abelian gauge fluctuations.

## 8.1 Chiral symmetry breaking mechanisms

In order to elucidate the chiral symmetry breaking mechanisms at work for colored fermions coupled to gravity, we analyze the system of beta functions in two distinct ways

a) We reduce the number of four-fermion channels in distinct ways, in order to discover whether there is a small set of channels that drives the system to criticality in Sec. 8.1.1.

b) We systematically remove structurally distinct terms from the beta functions to understand the impact of the combined as well as separate effect of gravitational and gauge-field fluctuations in Sec. 8.1.2.

We first focus on the case $N_{\mathrm{c}} = 3, N_{\mathrm{f}} = 6$ and investigate the system at other numbers of colors and flavors later.

### 8.1.1 Which channel drives the system to criticality?

We first consider $\lambda_{\pm}$ separately, i.e., the two channels which have the full symmetry of the kinetic term; we set the other two four-fermion couplings to zero. Neither gravity nor non-Abelian gauge fluctuations drive the system to criticality, i.e., the shifted Gaussian fixed point persists for all values of $G$ and $g$. Interestingly, this is different from the system with gravitational and Abelian gauge field fluctuations, where an upper bound on $e$ exists, cf. [11]. The difference between Abelian and non-Abelian fluctuations lies only in numerical prefactors (including signs) in the beta function, because the diagrams that contain gauge field fluctuations are the same for both types of gauge fields. These changes are sufficient to prevent chiral symmetry breaking in this subsector of the model.

Thus, we turn our attention to the subsystem spanned by $\lambda_{\mathrm{VA}}$ and $\lambda_{\sigma}$ and discover that chiral symmetry breaking occurs at $g_{\mathrm{crit}}(G)$. Conversely, there is also a critical value for $G$, which depends on $g$. The limits $g \to 0$ and $g_{\mathrm{crit}} \to 0$ are conceptually subtle, and we shall discuss these separately at the end of this section.

Finally, we can even discard the $\lambda_{\mathrm{VA}}$ channel and end up with a single channel, $\lambda_{\sigma}$, which drives the system to criticality. Its beta function, with all other four-fermion couplings and the Abelian gauge coupling set to zero, is

$$\beta_{\lambda_{\sigma}} = -\frac{57}{256\pi^2}g^4 - \frac{5}{8\pi}g^2 G + \left(2 - \frac{1}{6\pi}G - \frac{1}{\pi^2}g^2\right)\lambda_{\sigma} - \frac{3}{4\pi^2}\lambda_{\sigma}^2, \qquad (8.1)$$

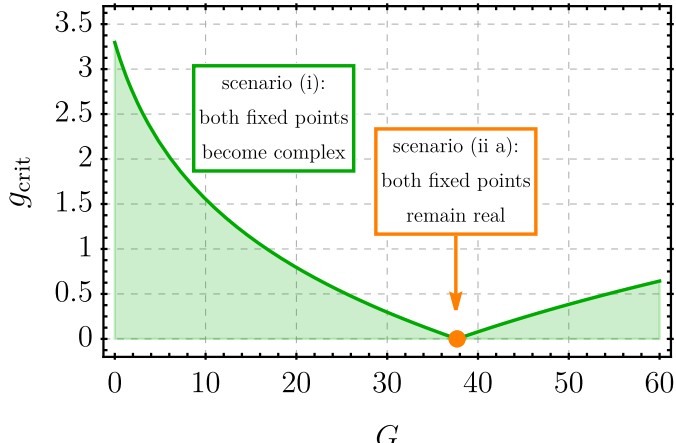

**Figure 9**. We plot $g_\text{crit}$ as a function of $G$ in the single channel approximation. The green line sets an upper bound on the value of $g$, about which the fixed point value $\lambda_\sigma$ becomes complex. The orange point is located at $G = 12\pi$. At this point, we have $g_\text{crit} = 0$.

at vanishing cosmological constant $\Lambda = 0$. The beta function is a downward-facing parabola, with a downward shift at increasing $g$ and $G$. Thus, the non-Abelian gauge interactions on its own as well as the combination of gravity and gauge fluctuations drive the system towards criticality. This is due to the terms $g^4$ and $g^2\,G$ in $\beta_{\lambda_\sigma}$, which encode how gravitational and gauge fluctuations first generate the coupling and then drive it to criticality.

The RG flow of the coupling is no longer bounded, when the $g^4$ and/or $g^2\,G$ term are large enough to trigger a fixed-point collision and subsequent complexification of the fixed points, i.e., scenario (i), which occurs at

$$g_\text{crit} = \sqrt{\frac{8\pi}{255}}\sqrt{74G + 192\pi - 3\sqrt{599G^2 + 3384\pi G + 2736\pi^2}}. \tag{8.2}$$

We now turn to the limit $g_\text{crit} \to 0$, which is a subtle limit and occurs at $G = 12\pi$. At this point, the fixed-point collision changes its character from scenario (i) to scenario (ii a), because at $g = 0$, the non-Abelian gauge symmetry becomes global. If one inverts $g_\text{crit}(G)$ to obtain $G_\text{crit}(g)$ (which is of course only possible in the range $G \in [0, 12\pi]$), one should again note that the limit $g \to 0$ is not a continuous one: since gauge field fluctuations decouple from the system in this limit, the fixed-point collision changes in character. Thus, if one were to ask for the critical value of $G$, beyond which no real fixed point exists, the answer at $g = 0$ would not be $G = 12\pi$, but instead $G \to \infty$. Instead, the continuous function $G_\text{crit}(g)$ actually tracks where a critical exponent becomes zero, because this property characterizes fixed-point collisions in scenario (i) and (ii).

This can also be seen by considering the imaginary part of the fixed-point value as a function of $G$ for varying values of $g$. For any non-zero value of $g$, the imaginary part is non-zero at $G = 12\pi$ and for increasing $g$, $\text{Im}(\lambda_\sigma)$ is symmetric about $G = 12\pi$. As $g$

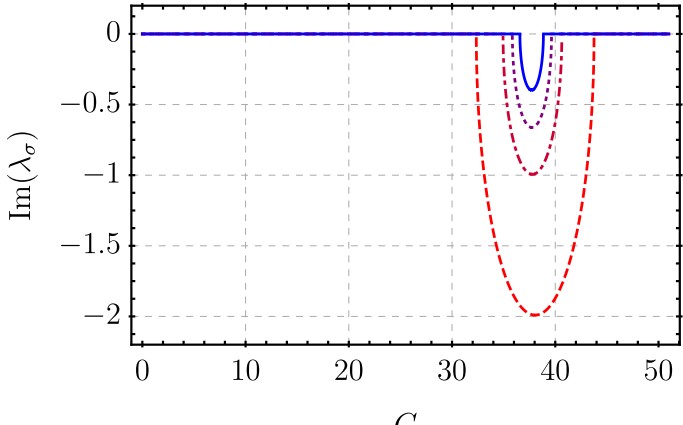

**Figure 10**. We show the imaginary part of $\lambda_\sigma$ as a function of $G$ for different values of the gauge coupling $g$. We vary $g$ from 0.04 (blue) to 0.2 (red).

decreases, the range of $G$ for which $\text{Im}(\lambda_\sigma) \neq 0$, shrinks to zero and vanishes in the limit $g \to 0$, cf. Fig. 10.

In Fig. 9 we plot $g_{\text{crit}}$ as a function of $G$. The value of $g_{\text{crit}}$ defines an upper bound on the non-Abelian gauge coupling if one wants to avoid chiral symmetry breaking. Our result shows that metric fluctuations reduces the value of $g_{\text{crit}}$, therefore strengthening the upper bound on non-Abelian gauge coupling at scales where gravity is dynamically important. For values $G \approx 1$, this works out to be $g_{\text{crit}} \approx 3$; for larger values of $G$, $g_{\text{crit}}$ tends towards zero and $g_{\text{crit}} = 0$ at $G = 12\pi$.

This provides a proof-of-principles that non-Abelian gauge couplings are in fact constrained in asymptotically safe quantum gravity. Reinstating the cosmological constant and inserting fixed-point values $G \approx 3.3$ and $\Lambda \approx -4.5$ (corresponding to the SM matter content, cf. [112]), we obtain $g_{\text{crit}} \approx 3.31$. We investigate the upper bound on the non-Abelian gauge coupling in more detail in Sec. 8.2 below.

### 8.1.2 What is the combined and the separate effect of gravitational and gauge-field fluctuations?

To determine the mechanism by which gravity and gauge fluctuations act together to break chiral symmetry, we investigate beta functions in which we artificially remove a subset of terms, namely those corresponding to either pure-gravity contributions or mixed gravity-gauge contributions.

Pure-gravity contributions occur only in $\beta_{\lambda_\pm}$ and are $\mathcal{O}(G^2)$. By removing them, we find virtually no effect on the value of $g_{\text{crit}}$ for $G \lesssim 3$, cf. left panel in Fig. 11; only at very large $G \gtrsim 10$ does $g_{\text{crit}}$ increase appreciably, if these terms are removed.

Mixed gravity-gauge contributions are $\mathcal{O}(g^2 G)$ and their removal strongly impacts $g_{\text{crit}}$ in the $\lambda_\sigma - \lambda_{\text{VA}}$ subspace, cf. right panel in Fig. 11. Because this is the subspace with the channels that drive the system to criticality, the removal of these terms also has a significant

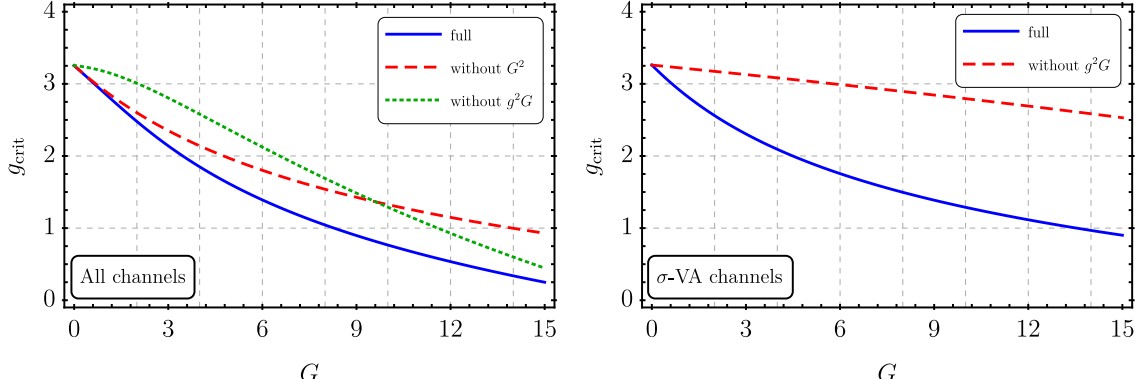

**Figure 11**. We show how different terms in the beta functions contribute to the mechanism of chiral symmetry breaking. The different lines correspond to $g_{\mathrm{crit}}$ obtained by neglecting specific terms in the beta functions. Left panel: results obtained with the full system. Right panel: results obtained with the two-channel approximation involving only $\lambda_\sigma$ and $\lambda_{\mathrm{VA}}$. In this approximation, the beta functions do not have terms proportional to $G^2$, as they are not induced by gravity.

effect on the full system, where $g_{\mathrm{crit}}$ increases and shows a less pronounced $G$-dependence for $G \lesssim 5$.

In conclusion, both sets of contributions, the pure-gravity and the gravity-gauge ones, favor chiral symmetry breaking, such that $g_{\mathrm{crit}}(G)$ remains a monotonically decreasing function, even if one set of contributions is removed.

## 8.2 Upper bound on the non-Abelian gauge coupling in asymptotically safe quantum gravity

For non-Abelian gauge couplings, the free fixed point $g_* = 0$ remains IR-repulsive even under the impact of quantum gravity; no interacting fixed point is induced. In the deep UV, the system with a non-Abelian gauge symmetry therefore has the same fixed-point structure as in [8]: the gauge coupling vanishes, so that $\lambda_{\mathrm{VA}}$ and $\lambda_\sigma$ can consistently be set to zero and gravity only induces $\lambda_\pm$.

Recall that for *Abelian* gauge couplings, the fixed-point structure is different, and induces an upper bound on the gauge coupling when coupled to asymptotically safe quantum gravity [113]. The upper bound arises from a competition between metric fluctuations and charged matter fluctuations. At leading (linear) order in the coupling, the antiscreening gravity fluctuations drive it to larger values; at the next (cubic) order in the coupling, the screening matter fluctuations drive it to smaller values; a balance is achieved at an asymptotically safe fixed point. This asymptotically safe fixed point constitutes the upper bound on the Planck-scale value of the coupling, because lower values are asymptotically free (due to the antiscreening effect of gravity, whereas larger values cannot be reached from a UV complete setting (due to the screening effect of matter resulting in a triviality problem). In contrast, because gauge bosons carry non-Abelian charge themselves and change the one-loop effect from screening to antiscreening, no such competition takes place

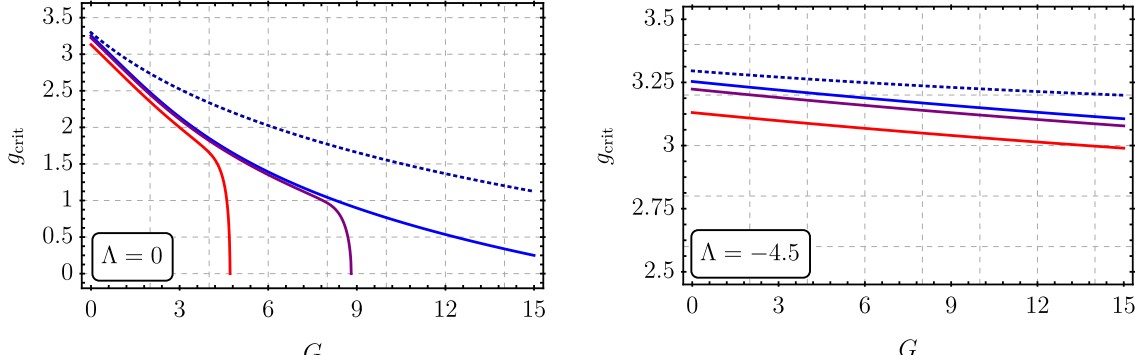

**Figure 12**. We plot $g_{\text{crit}}$ as a function of $G$ for several values of the Abelian gauge coupling, and for different values of the cosmological constant $\Lambda$. The blue, purple and red (solid) lines correspond to $e = 0$, $e = 0.5$ and $e = 1.0$, respectively. The dotted line correspond to the single channel approximation with $e = 0$.

for non-Abelian gauge couplings. Their values are thus a priori unbounded in asymptotically safe quantum gravity. This raises the question why both non-Abelian gauge couplings in the SM are perturbative and of order 0.5 at the Planck scale when, starting from their asymptotically free fixed point, they could grow without bounds without destroying the UV completeness of the matter-gravity theory.

Here, we have found, cf. Sec. 8.1, that the combined effect of gauge couplings and gravity drives chiral symmetry breaking in colored systems. We can use this fact to limit the value of non-Abelian gauge couplings in the asymptotically safe SM with gravity. We provide a preliminary estimate of this bound; a more detailed calculation requires us to account for the full flavor and color structure of the SM and the resulting proliferation of four-fermion couplings goes well beyond the scope of this paper.

We do so by following the shifted Gaussian fixed point in the full four-dimensional theory space of 4-Fermi couplings until it annihilates with another fixed point. Our numerical analysis, cf. Fig. 12, is in broad agreement with the simplified analytical study of Sec. 8.1, yielding $g_{\text{crit}} \approx 2.9$ at $G = 1$; $g_{\text{crit}}$ decreases, as $G$ increases.

In Fig. 12, we also show the impact of the Abelian gauge coupling on the value of $g_{\text{crit}}$. The bound is strengthened in the presence of the Abelian gauge coupling. For example, setting $e = 0.5$, we obtain $g_{\text{crit}} \approx 2.8$ at $G = 1$.

To summarize, we see that gravity, Abelian and non-Abelian gauge fluctuations all act together to trigger chiral symmetry breaking, cf. Fig. 12. This is distinct from the analysis in [8], where gravity protects fermions from acquiring masses. The cases of colorless and colored fermions are thus fundamentally distinct from each other, because more channels become available, in which gravity acts differently than in the simpler system without color and charge.

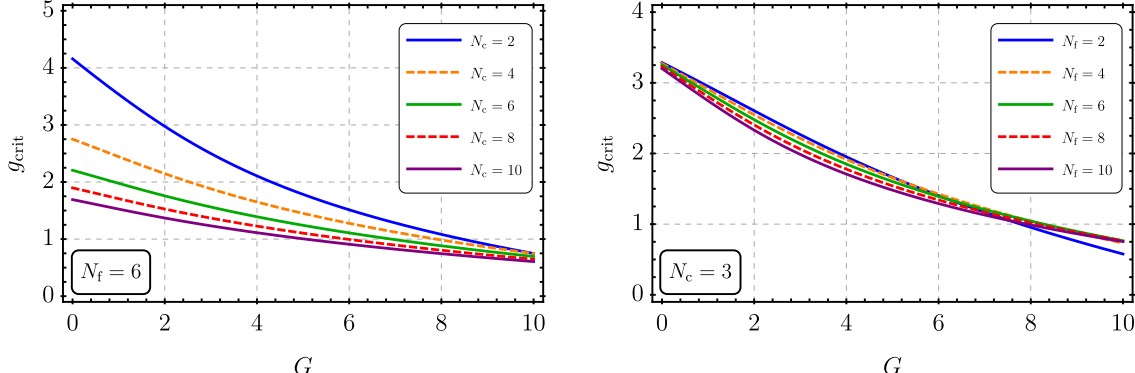

**Figure 13**. We plot $g_{\text{crit}}$ as a function of $G$ for different values of $N_{\text{f}}$ and $N_{\text{c}}$. As we see, changes on $N_{\text{f}}$ and $N_{\text{c}}$ has a quantitative impact on $g_{\text{crit}}$, but the qualitative aspects of our results remains unchanged.

### 8.3 Beyond the Standard Model: chiral symmetry breaking away from $N_{\text{c}} = 3, N_{\text{f}} = 6$.

Because $N_{\text{f}}$ and $N_{\text{c}}$ appear in the beta functions of the system, it is an obvious question whether our results depend qualitatively on these two parameters. We investigate this by calculating $g_{\text{crit}}(G)$ for various choices of $N_{\text{f}}$ and $N_{\text{c}}$. First, we find that there is a very mild dependence on $N_{\text{f}}$ only, cf. right panel of Fig. 13; this holds also away from $N_{\text{c}} = 3$. Conversely, there is a stronger dependence on $N_{\text{c}}$, cf. left panel of Fig. 13. The higher $N_{\text{c}}$, the lower $g_{\text{crit}}$, which may be understood from the fact that there are $N_{\text{c}}^2 - 1$ non-Abelian gauge bosons and thus their effect grows with $N_{\text{c}}$. We do, however, find that, at least within the range of $N_{\text{f}}$ and $N_{\text{c}}$ that we considered, no qualitative differences arise to the case $N_{\text{c}} = 3$ and $N_{\text{f}} = 6$. From an inspection of the beta functions, it is also not to be expected that qualitative differences arise.

This result paves the way for model-building beyond the Standard Model, e.g., in a context inspired by composite-Higgs-like settings for the dark and/or visible sector.

### 8.4 Lower bounds on the number of fermions

Next, we explore the effect of color on the lower bound on the number of flavors that was found in [11] for a system of $N_{\text{f}}$ fermions charged under an Abelian gauge group. We assume that all $N_{\text{f}} \cdot N_{\text{c}}$ fermions carry the same charge under the Abelian gauge group. We proceed in analogy to [11] (and correct a numerical error in that paper) and supplement the system of RG equations with the beta function for the Abelian gauge coupling,

$$\beta_e = \frac{N_{\text{f}} \cdot N_{\text{c}}}{12\pi^2} e^3 - f_g \, e, \tag{8.3}$$

at one loop, with the gravitational contribution

$$f_g = \frac{5\,G}{9\pi(1 - 2\Lambda)} - \frac{5\,G}{18\pi(1 - 2\Lambda)^2} \,. \tag{8.4}$$

This results in an asymptotically safe fixed point at

$$e_* = \sqrt{\frac{12\pi^2 f_g}{N_{\mathrm{f}} \cdot N_{\mathrm{c}}}}. \tag{8.5}$$

An increasing number of colors and flavors lowers the fixed-point value for the gauge coupling; thus, a lower bound on the number of charged fermions follows, because one needs $e_* < e_{\mathrm{crit}}$ to avoid chiral symmetry breaking induced by gravity, with $e_{\mathrm{crit}}$ the critical value for chiral symmetry breaking. In the presence of a non-Abelian gauge group, the bound gets strengthened: while the fixed-point value for $g$ is zero, $g$ increases towards the Planck scale. The combined effect of gravity, Abelian and non-Abelian gauge interactions must be low enough to avoid chiral symmetry breaking in all four channels. The two new ingredients compared to [11] thus are the existence of two new channels as well as the non-Abelian gauge contribution to all four channels.

We first perform a single-channel analysis, focusing again on $\lambda_\sigma$. In this approximation, we find the shifted Gaussian fixed point

$$
\begin{aligned}
\lambda_{\sigma,*} = {} & \frac{4\pi^2}{N_{\mathrm{c}}} - \frac{3\,e^2}{2\,N_{\mathrm{c}}} + \frac{3\,(1 - N_{\mathrm{c}}^2)\,g^2}{4N_{\mathrm{c}}^2} - \frac{G}{4N_{\mathrm{c}}}\left(\frac{5\pi}{(1 - 2\Lambda)^2} - \frac{135\pi}{9 - 12\Lambda^2} - \frac{18\pi}{(9 - 12\Lambda)}\right) \\
& + \frac{2\pi^2}{N_{\mathrm{c}}}\left\{\left[2 - \frac{3\,e^2}{4\pi^2} + \frac{4\,(1 - N_{\mathrm{c}}^2)\,g^2}{8\pi^2\,N_{\mathrm{c}}} - \frac{G}{8\pi^2}\left(\frac{5\pi}{(1 - 2\Lambda)^2} - \frac{135\pi}{9 - 12\Lambda^2} - \frac{18\pi}{(9 - 12\Lambda)}\right)\right]^2 \right. \\
& - \frac{N_{\mathrm{c}}\,g^2}{\pi^2}\left[\frac{9\,e^2}{16\pi^2} - \frac{9(8 - 3N_{\mathrm{c}}^2)\,g^2}{256\pi^2 N_{\mathrm{c}}^2}\right. \\
& \left.\left. + \frac{G}{160\pi}\left(\frac{50}{(1 - 2\Lambda)^2} + \frac{50}{(1 - 2\Lambda)} - \frac{243}{(9 - 12\Lambda)^2} + \frac{27}{(9 - 12\Lambda)}\right)\right]\right\}^{1/2}. \tag{8.6}
\end{aligned}
$$

We note that $\lambda_{\sigma,*}$ is independent of $N_{\mathrm{f}}$ in the single-channel approximation. Accordingly, it gives rise to a critical value of $e$ that depends on $G, \Lambda, g$ and $N_{\mathrm{c}}$, only. Comparing to the fixed-point value for $e$ and setting $G, \Lambda$ to their fixed-point values, we find no lower bound on $N_{\mathrm{c}}$; the non-Abelian gauge group thus remains unconstrained, as does the number of fermions, cf. Fig. 14.

Next, we turn to the analysis with all four-fermion channels. In this case, the fixed-point structure depends on $N_{\mathrm{f}}$ and $N_{\mathrm{c}}$, in contrast to the abovementioned single-channel approximation. However, the qualitative result obtained with all channels is similar to what we have obtained within the single-channel approximation, cf. Fig. 14. For $g = 1/2$, which is of similar magnitude as the Planck-scale value of both non-Abelian gauge couplings in the Standard Model, we find that the critical line defined by $e_{\mathrm{crit}}(N_{\mathrm{c}})$ (with fixed $N_{\mathrm{f}}$) lies above the upper bound on the Abelian gauge coupling established by asymptotically safe quantum gravity. Thus, within asymptotically safe quantum gravity, fermions are protected from chiral symmetry breaking triggered by a combination of gravitational and gauge field fluctuations.

In Fig. 14, we also show $e_{\mathrm{crit}}(N_{\mathrm{c}})$ for $g = 1$, $g = 2$ and $g = 2.5$. Increasing the value of $g$ lowers the line corresponding to $e_{\mathrm{crit}}(N_{\mathrm{c}})$. Thus, for sufficiently large values of $g$, the

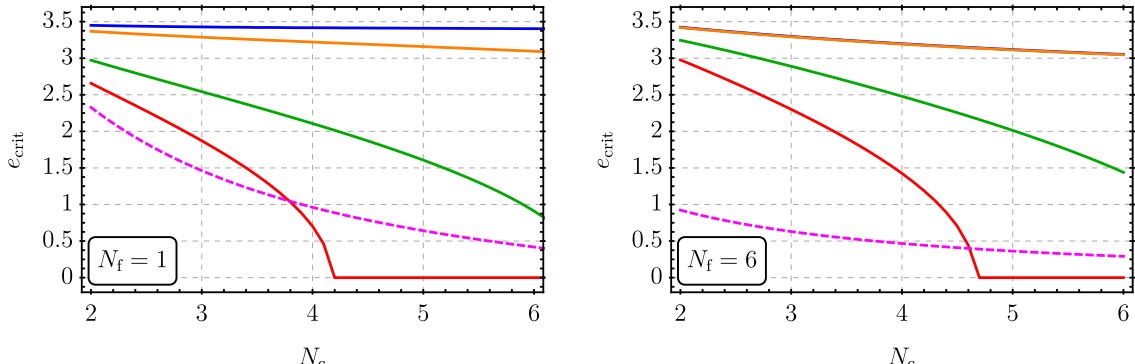

**Figure 14**. We show $e_{\text{crit}}$ as a function of $N_c$ for several values of the non-Abelian gauge coupling $g$. The (solid) lines in blue, orange, green and red correspond to $g = 0.5$, $g = 1.0$, $g = 2.0$ and $g = 2.5$, respectively. The magenta (dashed) line indicates the fixed-point value $e_*$ as a function of $N_c$. In all cases, we set the gravitational couplings $G$ and $\Lambda$ to their fixed-point values.

upper bound on the Abelian gauge coupling is not sufficient to protect fermions from chiral symmetry breaking.

### 8.5 Non-Abelian gauge coupling with asymptotically safe UV-completion

In this section, we discuss a different scenario for UV completion in the non-Abelian gauge sector. For small enough flavor number, the UV completion of non-Abelian couplings relies on asymptotic freedom. However, asymptotic freedom may be spoiled beyond the Standard Model if one adds too many fermions charged under the non-Abelian gauge group. Here, we explore this possibility, but with the inclusion of graviton fluctuations.

For a non-Abelian gauge field coupled to $N_f$ fermions and gravity, we obtain the following one-loop beta function for the gauge coupling $g$:

$$\beta_g = -\frac{1}{16\pi^2}\left(\frac{11}{3}N_c - \frac{2}{3}N_f\right)g^3 - f_g\,g, \tag{8.7}$$

with $f_g$ being the gravitational contribution [c.f., Eq. (8.4)]. If one discards the gravitational contribution and the number of flavors exceeds the upper bound $N_f = 11\,N_c/2$, one loses asymptotic freedom, and the non-Abelian gauge sector suffers from a UV Landau pole similar to the Abelian one.

In the presence of gravity, one can achieve a UV completion even if the number of flavors exceeds the upper bound $N_f = 11\,N_c/2$, as long as the gravitational contribution acts with an anti-screening effect (i.e., $f_g > 0$). This has also been explored previously in the context of Grand Unified Theories [144] and gauge theories which are asymptotically safe even without gravity [145]. In the scenario with $N_f > 11\,N_c/2$, the one-loop beta function for the non-Abelian gauge coupling admits an interacting fixed point with

$$g_* = \sqrt{\frac{48\pi^2\,f_g}{2N_f - 11N_c}}\,. \tag{8.8}$$

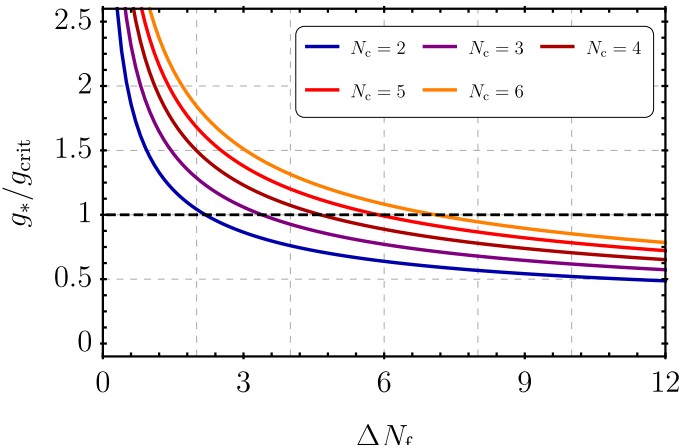

**Figure 15**. We plot the ratio between the fixed point value $g_*$ (c.f. Eq. (8.10)) and $g_{\mathrm{crit}}$ as a function of $\Delta N_{\mathrm{f}}$, as defined in Eq. (8.9), for different values of $N_{\mathrm{c}}$. We set the gravitational couplings to $G = 1$ and $\Lambda = 0$.

This fixed point is IR-attractive and bounds the (trans-)Planckian value of $g$ from above.

Next, we explore the possibility of chiral symmetry breaking in the scenario associated with the fixed point of Eq. (8.8). First, we note that if $N_{\mathrm{f}}$ approaches the value $11N_{\mathrm{c}}/2$ from above, the fixed-point value $g_*$ increases. Thus, it is conceivable that if $N_{\mathrm{f}}$ is just slightly above $11N_{\mathrm{c}}/2$, then $g_*$ will be larger than $g_{\mathrm{crit}}$, which is the upper bound to avoid chiral symmetry breaking. In this case, it is useful to introduce a new variable

$$\Delta N_{\mathrm{f}} = N_{\mathrm{f}} - \frac{11}{2} N_{\mathrm{c}} \,, \tag{8.9}$$

which allows us to rewrite $g_*$ according to

$$g_* = \sqrt{\frac{24\pi^2 f_g}{\Delta N_{\mathrm{f}}}} \,. \tag{8.10}$$

In Fig. 15, we plot the ratio $g_*/g_{\mathrm{crit}}$ as a function of $\Delta N_{\mathrm{f}}$ for various choices of $N_{\mathrm{c}}$. In all cases, we set $G = 1$ and $\Lambda = 0$ as representative values for the gravitational couplings.[11] To avoid chiral symmetry breaking in the fixed-point regime associated with $g_*$, one requires $g_*/g_{\mathrm{crit}} < 1$. From Fig. 15, one can see that this requirement defines a lower bound on $\Delta N_{\mathrm{f}}$ as a function of $N_{\mathrm{c}}$. For fixed values of $G$ and $\Lambda$, we find that the lower bound on $\Delta N_{\mathrm{f}}$ increases with $N_{\mathrm{c}}$.

We note that $g_*$ (cf. (8.8)) only defines an upper bound on the non-Abelian gauge coupling. The beta function (8.7) also admits asymptotically free trajectories starting from $g_{*,\mathrm{free}} = 0$ in the UV. Therefore, our analysis for the ratio $g_*/g_{\mathrm{crit}}$ focuses on the maximally

---

[11]In a more consistent treatment, we should use the fixed-point values of $G$ and $\Lambda$. However, using $\beta_G$ and $\beta_\Lambda$ obtained via background approximation, we cannot find real fixed points if $N_{\mathrm{f}} > 11N_{\mathrm{c}}/2$. Fluctuation calculations indicate that it should be possible to accommodate a large number of fermions [90, 106], thus we use representative values for $G$ and $\Lambda$ to give a proof of principle.

predictive scenario where a UV-completion is realized in terms of the interacting fixed point $g_*$. Thus, it is more appropriate to read the lower bound on $\Delta N_\mathrm{f}$ as a value that allows one to safely combine avoided chiral symmetry breaking in the (trans-) Planckian regime with an asymptotically safe prediction of the Planck-scale value of the non-Abelian gauge coupling.

## 9    Conclusions and outlook

In the present paper, we have investigated whether fermions can remain light under the impact of quantum gravity fluctuations if they carry both color and Abelian charge. The answer is positive, at least if the gravitational coupling and the gauge couplings are small enough. All couplings are bounded from above by the requirement of light fermions. For the non-Abelian gauge coupling, this is the first indication that it is bounded from above in asymptotically safe gravity.

At a more formal level, our system serves as a case study for the different types of fixed-point collisions that exist:

1.) First, as long as the non-Abelian symmetry is global, fixed-point collisions proceed along the real axis in the space of couplings. Fixed points trade stability at a critical value of $G$, but a fully stable fixed point, i.e., one in which all four-fermion couplings are irrelevant, always exists. Therefore, gravitational fluctuations on their own cannot break chiral symmetry, even if fermions are charged under both an Abelian and a non-Abelian global symmetry group.

2.) Second, when the non-Abelian symmetry is local, fixed-point collisions result in complex conjugate pairs of fixed points. Then, the fully stable fixed point is not available for all values of the couplings, instead, there is a critical line for the non-Abelian gauge coupling $g$, as a function of the gravitational coupling $G$, i.e., $g_\mathrm{crit} = g_\mathrm{crit}(G)$ above which chiral symmetry breaks. Gravitational and gauge field fluctuations act together to trigger chiral symmetry, such that $g_\mathrm{crit}(G)$ is a decreasing function.

Our study is also an example of the interplay of gravitational fluctuations with global symmetries: in the case with a global non-Abelian symmetry, gravitational fluctuations do not only preserve both the $\mathrm{U}(N_\mathrm{c})$ global color symmetry and $\mathrm{U}(N_\mathrm{f})_\mathrm{L} \times \mathrm{U}(N_\mathrm{f})_\mathrm{R}$ flavor symmetries, but even enhance it: at $G < G_\mathrm{crit}$, it is characterized by an enhanced global $\mathrm{U}(N_\mathrm{f} N_\mathrm{c})_\mathrm{L} \times \mathrm{U}(N_\mathrm{f} N_\mathrm{c})_\mathrm{R}$ symmetry, whereas at $G > G_\mathrm{crit}$, it shows a smaller enhanced $\mathrm{U}(N_\mathrm{f})_\mathrm{L} \times \mathrm{U}(N_\mathrm{f})_\mathrm{R} \times (U(N_\mathrm{c})_\mathrm{L} \times \mathrm{U}(N_\mathrm{c})_\mathrm{R})^{N_\mathrm{f}}$ symmetry. We thus find that even at large $G$, there is always some symmetry enhancement above the minimal $\mathrm{U}(N_\mathrm{f})_\mathrm{L} \times \mathrm{U}(N_\mathrm{f})_\mathrm{R} \times \mathrm{U}(N_\mathrm{c})$ symmetry. This is in contrast with the no-global-symmetries swampland conjecture in quantum gravity [146, 147], which states that, due to black-hole configurations in the path integral, global symmetries should always be broken when metric fluctuations are integrated over. That singular black-hole configurations interfere destructively in the path integral when interactions present an asymptotically safe dynamics are assumed [148, 149] hints at asymptotic safety being a potential counterexample to this conjecture. Overall, our

study is another example hinting at major differences between string-inspired swampland conjectures and asymptotically safe results regarding global symmetries. On the one hand, the stringy swampland may not contain global continuous symmetries at all [150], and even for local symmetries, the couplings may not be too small [151]. On the other hand, in asymptotic safety, numerous examples of calculations exist in which quantum gravity fluctuations preserve, or, as in the present case, even enhance global symmetries, and the values of gauge couplings can be arbitrarily small and may even be bounded from above, see [87] for a summary and critical discussion of these results.

**Outlook: Towards technicolor and composite Higgs sectors in asymptotically safe gravity**  Our study provides the main ingredients necessary to investigate technicolor-like or composite Higgs sectors within asymptotically safe gravity. Such sectors are based on strongly-coupled fermion sectors in which chiral symmetry breaking results in bound-state formation. If quantum gravity would not leave chiral symmetry intact, then such sectors would generically be incompatible with asymptotic safety, because the desired mass scale of the bound states is typically close to LHC scales, or at least considerably below the Planck scale. Given our result, a compatibility between technicolor and composite Higgs models and asymptotic safety seems in principle viable and deserves further study, both in the context of such models as an extended Higgs sector and for dark matter beyond the Standard Model.

**Outlook: Towards light fermions in the Standard Model**  Our study also paves the way towards testing whether the Standard Model (SM) fermions can remain light in asymptotically safe quantum gravity. However, there are several structural aspects to take into account: first, the Abelian hypercharge coupling of different flavors within one generation is different, such that different prefactors need to be introduced in the minimal coupling to the Abelian gauge field. Further, the non-Abelian SU(2) gauge fields only couple to the left-handed components of the fermions, requiring us to either work with Weyl fermions or to introduce the appropriate projection operators. This will result in some modifications of the beta functions.

Further, distinguishing all 6 different flavors of quarks and 6 different flavors of leptons in the SM, there are 2751 four-fermion operators [129, 152]. These arise from 25 different structures [152], which give rise to 2751 different couplings because of the many ways in which flavor indices within each structure can be contracted. This number of operators is clearly prohibitively large; thus, a computation needs to be combined with physical insight to reduce the number of couplings. The difference between the generations is driven by the Yukawa couplings, thus, we expect all four-fermion operators that involve the top quark and are driven by the top-Yukawa coupling, to be parametrically enhanced compared to the other four-fermion operators. A feasible upgrade of our study towards light Standard Model fermions thus consists in singling out those four-fermion operators which include top quarks and thereby reduce the number of independent four-fermion operators to be considered to a tractable number.

## Acknowledgments

This work is supported by a research grant (29405) from VILLUM FONDEN. S. R. further acknowledges support from the Deutsche Forschungsgemeinschaft (DFG) through the Walter Benjamin program (RA3854/1-1, Project id No. 518075237).

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
