# Peer review of "Light fermions in color: why the quark mass is not the Planck mass"

_SciPost Physics_

## Round 1 · Referee Report · Anonymous (Referee 1) · 2024-3-25

Report

The theme of this paper, most broadly put, is: Can the hypothesis of asymptotic safe gravity be consistent with the experimental fact that the electroweak symmetry breaking scale is much lower than the Planck scale? Or more specifically, as being asked in this paper, can asymptotically safe gravity lead to a condensate of SM fermions that breaks EW symmetry at the Planck scale? If it can, what is the range of couplings in which such catastrophe is avoided?

The question is important in the framework of asymptotically safe gravity. The concern comes directly from the very assumption of asymptotically safe gravity, namely, the existence of an interacting UV fixed point in the space of couplings including gravitational couplings. Since gravitational interactions are irrelevant interactions around the trivial fixed point, we must accept that at the nontrivial fixed point the theory has to be strongly coupled. At least gravity for sure gets strong at the Planck scale simply due to its canonical scaling dimension. One might say that is only a weak coupling statement and who knows what the actual scaling dimension is at strong coupling, but then he/she has already bought the proposition that it is strongly coupled. Once gravity is strong, some other couplings may also get strong, and it would be problematic if such strong interactions lead to a condensate of SM fermions that breaks EW symmetry at the Planck scale (in a somewhat analogous way as the formation of q-qbar condensate by QCD that would break EW symmetry even in the absence of the Higgs VEV, although this one is due to a relevant interaction getting strong in the IR as opposed to an irrelevant interaction going strong in the UV). Therefore, the problem studied in this paper is a must-ask question in the framework of asymptotically safe gravity simply because of the very nature of this hypothesis. Toward this goal the paper aims to study the question in a somewhat simpler theory than the SM, where they look at vectorlike fermions charged under a gauged U(1) or SU(N) and study the fate of the chiral symmetry of the fermions in the UV.

What I am really having trouble with is that their analysis assumes that the anomalous dimensions are small and the scaling dimensions take near-canonical values. I understand that some approximation/truncation scheme is practically needed to study a strongly coupled problem, but the one approximation we don't want to make is that the couplings are weak! The beta functions presented in the paper have the form of "canonical scaling + 1-loop corrections", but the perturbative corrections cannot offset canonically irrelevant dimensions by O(1) to make them marginal, unless the couplings are large. But once the couplings are so large that 1-loop anomalous dimensions can balance against canonical (i.e., tree-level) scalings, the 1-loop calculations do not tell us anything quantitatively meaningful as 2-loop and higher-loop corrections should be just as important. The detailed numerical coefficients in the beta functions and the detailed numerical results as presented in this paper are not justified in the sense that they give the false impression that the results are quantitatively under control, whereas every term in their beta functions is at best only a very rough estimate (at the level of Naive Dimensional Analysis) modulo an unknown factor and even an unknown sign. The issue is somewhat analogous to keeping more significant digits in experimental measurements than justified and drawing conclusions based on those extra digits.

Therefore, I find the claims made by the authors in the paper too aggressive as I don't see a calculational framework by which such quantitative statements can be made.

---

## Round 1 · Referee Report · Anonymous (Referee 2) · 2024-5-11

Report

In this paper, the author studies chiral symmetry breaking in the presence of abelian/non-abelian gauge and gravitational interactions coupled to fermions with flavor structures. In particular, for the gravitational interaction, the asymptotic safety scenario is assumed. The authors investigate the impact of gauge and gravitational interactions on the four-fermion interactions and discuss whether these interactions trigger chiral symmetry breaking or not. If chiral symmetry breaking takes place at high scale, e.g., at the Planck scale, fermions such as quarks and leptons may acquires large masses proportional to the breaking scale. This becomes a serious obstacle for formulating asymptotically safe gravity. Conversely, this provides constrains on model setups. The authors conclude that fermions remain light for large gravitational (Newton) coupling values when the non-Abelian gauge symmetry is global, while when it is localized, the chiral symmetry is broken for a larger value of gauge/gravitational couplings than the critical coupling.

The motivation of this work is clear and the obtained results are interesting. However, I have several questions and suggestions. I wish the authors address them before making my recommendation.

(i) Chiral symmetry is a kind of flavor symmetry. Therefore, the authors should note before Eq.(3.4) what kind of flavor symmetry is assumed. U(N_f)_R × U(N_f)_L or U(1)_V × SU(N_f)_R × SU(N_f)_L?

(ii) The importance of the Fierz complete basis is discussed in Refs. [J.Jaeckel and C.Wetterich, Phys. Rev. D 68 (2003), 025020]. This may support the statement below Eq. (3.5).

(iii) The authors utilize four-fermion couplings to monitor the occurrence of chiral symmetry breaking. However, this may lead to misinterpretation. The divergence of the four-fermion interactions does not invariably indicate the onset of the chiral symmetry breaking. These couplings are related to chiral susceptibility and the correlation length, potentially suggesting a phase transition upon their divergence. Specifically, in the context of a second-order phase transition, such divergence signifies chiral symmetry breaking, yielding a non-zero order parameter, known as the chiral condensate <bar ψψ>.
However, we have to discuss carefully the case of a first-order phase transition. In the language of bosonization, for instance, the inverse four-fermion coupling in the scalar-pseudoscalar channel, denoted as λ_σ, correlates with the mass-squared parameter of the sigma meson field σ ~ <barψψ>. Consequently, the divergence of λ_σ corresponds to the zero curvature of the effective potential at its origin. Even if the four-fermion interactions do not diverge, the symmetric vacuum may become meta-stable and there could exist a stable vacuum at which the symmetry is broken.
Notably, the order of the phase transition can depend on the flavor number and current fermion masses, a concept depicted in the Columbia plot. In scenarios with massless many flavor fermions, a first-order phase transition might manifest more prominently (although this is still an open question). Consequently, one cannot definitely conclude chiral symmetry breaking does not occur due to the gravitational interaction.
The authors should note this fact, and mention that in this paper, the second-order chiral phase transition is always assumed.

(iv) In the presence of the gauge and gravitational fields, U(1)_A symmetry may be broken by the instanton (or anomaly) effect as argued by ’t Hooft [Phys. Rev. Lett. 37 (1976), 8-11; Phys. Rept. 142 (1986), 357-387]. Why do the authors not consider such a case? In the asymptotic safety scenario, the anomalous breaking of U(1)_A due to the gravitational instanton is discussed in Ref. [Y. Hamada, et al, Phys. Rev. D 103 (2021) no.10, 106016]. In the strongly interacting regime, the gauge and gravitational interactions could potentially induce the instanton configuration for which the nonzero Dirac index is given. Thus, not only the “gravitational coupling” effects trigger chiral symmetry breaking, but also non-trivial configurations (solitons) could contribute to the symmetry breaking effect. Indeed, the recent lattice QCD [S.~Aoki et al., PTEP 2022 (2022) no.2, 023B05] indicates the U(1)_A chiral symmetry breaking is triggered by instanton and then this breaking effect triggers SU(N_f)_A.
Hence, one cannot always conclude that the gravitational interactions do not break the chiral symmetry even if the “gravitational coupling” effects on the four-fermion couplings do not trigger the symmetry breaking. This fact should be noted as well as the above statement (iii). Or, if there is a clear reason why U(1)_A symmetry breaking is not necessarily taken into account, the authors should argue in the paper.
(Note that this anomaly is different from the gauge anomaly which becomes an obstacle to formulated a gauge theory and is mentioned in the beginning of Section 2.)

(v) Why do abelian and non-abelian gauge couplings not contribute to the fermion anomalous dimension η_ψ? Is it due to an approximation? In the usual perturbative calculation, the fermion anomalous dimension is induced by these gauge fields.

Recommendation

Ask for major revision

---

## Editorial Decision

awaiting_resubmission